# LISTENING TO THE WISE FEW: SELECT-AND-COPY ATTENTION HEADS FOR MULTIPLE-CHOICE QA

## ABSTRACT

Multiple-choice question answering (MCQA) is one of the most widely adopted methods for evaluating large language models (LLMs). In this approach, the model is presented with a question and a set of possible answers, and the answer with the highest logit is selected as the model's prediction. However, this evaluation format has limitations, as even if the model knows the correct answer, it may struggle to select the corresponding option simply due to difficulties in following this rigid format. Methods such as instruction tuning or in-context learning help alleviate this issue but introduce their own biases, such as dependence on the order and semantics of training examples. In this paper, we address this issue by conducting an intrinsic investigation of the LLM's decision-making process when answering multiple-choice questions. Specifically, we identify and study specific *select-and-copy* heads responsible for choosing the correct answer. We develop new scores to reveal the underlying knowledge from these heads: the Query-Key Score, which measures the interaction between query and key representations in the selected head, and the Attention Score, which is based on the attention weights. By studying these scores, we found that the most pronounced *select-and-copy* heads are consistent across four popular Multi-Choice Question Answering (MCQA) datasets. Moreover, our scores enable better knowledge extraction, achieving up to a 16% gain for LLaMA2-7B and up to 10% for larger models on these benchmarks. On a synthetic dataset, where the correct answer is known explicitly, accuracy increases by nearly 60%, confirming the method's effectiveness in overcoming MCQA format limitations. To support our claims, we conduct experiments on models ranging from 1.5 billion to 70 billion parameters, in both zero-shot and few-shot settings.

## 1 INTRODUCTION

Questions with multiple answer options are a common form of benchmarks for evaluating question answering (Hendrycks et al., 2021), common sense (Zellers et al., 2019), reading comprehension (Huang et al., 2019), and other abilities of Large Language Models (LLMs). In multiple choice question answering tasks (MCQA), the model is provided with the question and multiple answer options, e.g. *"Question: How many natural satellites does the Earth have? Options: A. 0. B. 1. C. 2. D. 3. E. I don't know. F. None of the above."* Sometimes context, such as a paragraph or dialogue, may be provided to help answer the question, particularly for reading comprehension or common sense reasoning tasks. The model is then expected to select the letter corresponding to the correct answer. This format resembles real-life student exams and offers the benefit of straightforward evaluation using automated tools.

On the other hand, for LLMs, especially smaller ones, understanding and adhering to this format is not always trivial. Their performance on a given multiple-choice dataset depends not only on their ability to solve the task but also on in-context learning and instruction-following capabilities. They may provide correct answers with formatting issues, complicating automatic evaluation. As a result, some studies delegate answer evaluation to another LLM instead of relying on exact string comparison (Wang et al., 2024). Besides, when assessing the logits of a model for multiple-choice options, LLMs can sometimes follow shallow patterns, such as the distribution of options. For example, some models tend to favor option "A" while others are more inclined to choose "D" (Zheng et al., 2024a). These issues highlight the pitfalls of the current MCQA evaluation process.

Figure 1: Our method calculates the Query-Key score between the end-of-line token of an answer option and the last token of the prompt for the designated head, from which we derive the answer.

However, the model's inability to follow the task format does not mean it lacks knowledge of the correct answer. This work shows that while small LLMs may struggle on MCQA benchmarks, their intermediate attention states provide valuable insights. Specifically, we identify *select-and-copy* heads that choose semantically relevant options and propagate their representations, and we introduce a whilte-box method that uses the queries and keys from these heads to select correct answers. In particular, our findings suggest that LLMs process MCQA tasks better in the middle layers, but later layers tend to revise this information, sometimes reducing performance.

We identify the key algorithmic operation performed by pretrained Transformer models when answering multiple-choice questions. This task involves the model first computing a representation of the semantic information in both the question and the options within specific attention heads. The model then selects the most appropriate option using the query-key alignment mechanism (see section 3), followed by copying and outputting the option. Based on this, we identify the heads that perform this *select-and-copy* operation on the aggregated embeddings of the possible answers. Our results show that such heads exist in all models we tested, from 1.5 billion to 70 billion parameters. Notably, the best-performing heads are consistent across datasets, and the answers generated by these heads are often more accurate than the model's final output, particularly in zero-shot scenarios.

Our contributions are as follows: (1) We show that select-and-copy heads are present in LLMs ranging from 1.5B to 70B parameters, performing the option selection operation for MCQA tasks, and the best of them are consistent across datasets; (2) We introduce the QK-score and attention score, a novel option-scoring methods based on key-query representations from these heads, which improve accuracy by 9-16% with task-specific heads; (3) We demonstrate that our method is more stable than baselines when handling option permutations and the addition of supplementary options (e.g., "I don't know"); (4) Our results support the hypothesis that semantic representations are encoded in specific heads in the query, key, and value vectors of a phrase's final tokens (namely, end-of-sentence or end-of-line tokens), as observed in previous studies (e.g., Li et al. (2023b); Stolfo et al. (2023)); (5) We analyze the attention patterns of *select-and-copy* heads and their behavior under different conditions, advancing our understanding of how attention mechanism can work as select and copy operation and how LLMs function in general.

## 2 RELATED WORK

Question-answering datasets are commonly used to assess the capabilities of Large Language Models (LLMs) in terms of knowledge retention, text comprehension, and reasoning abilities. The results of such evaluations are reported in numerous technical papers on recent LLMs, including LLaMA2 and LLaMA3, GPT-4, and Claude 3 Opus (et. al., 2023; Dubey et al., 2024; OpenAI, 2024; Anthropic, 2024). Additionally, multiple-choice question answering (MCQA) tasks are frequently included in benchmark tests for these models, as they provide a straightforward method for evaluation (Ye et al., 2024; Pal et al., 2022).

There are several approaches to the multiple-choice question answering (MCQA) task. One common approach, multiple-choice prompting (MCP), involves presenting the model with a question and multiple answer options at once. Despite recent studies highlighting issues such as answer order bias (Gupta et al., 2024; Pezeshkpour & Hruschka, 2024) and selection bias (Zheng et al., 2024a), MCP offers several advantages over cloze prompting (CP), where the model is presented with the

question and only one option at a time (Robinson & Wingate, 2023). In CP, normalized answer probabilities are typically used for evaluation, while in MCP, the probabilities of the individual tokens in the answer options serve as a proxy for evaluation.

In this work, we investigate the inner mechanisms of large language models, with a particular focus on the role of attention heads in multiple-choice question answering tasks. The functional roles of attention heads have been analyzed for transformer-based models since the early development of encoder-only architectures (Jo & Myaeng, 2020; Pande et al., 2021). More recently, even more detailed approaches have been developed for decoder-only models as part of the broader field of mechanistic interpretability (Elhage et al., 2021; Olsson et al., 2022; Bricken et al., 2023). For example, *induction heads*, identified by Elhage et al. (2021), are crucial for in-context learning (Olsson et al., 2022; Von Oswald et al., 2023), indirect object identification (Wang et al., 2023), and over-thinking (Halawi et al., 2024). Additionally, several studies have linked theoretically constructed networks with real pretrained language models, revealing various types of attention heads, such as constant heads (Lieberum et al., 2023), negative heads (Yu et al., 2024), and content gatherer heads (Merullo et al., 2024). For further details on mechanistic interpretability and the role of attention heads, we refer readers to Rai et al. (2024) and Zheng et al. (2024b).

We focus on *select-and-copy heads*, which are used to select the correct option in the MCQA task. A special case of these heads, which focuses on option labels, was discussed in (Lieberum et al., 2023). However, we demonstrate that similar heads, when focusing on other tokens, achieve higher accuracy compared to the final model output.

Moreover, our experiments show that the heads that outperform the baseline on MCQA are located in the middle layers of the LLM. This finding aligns with previous research suggesting that while much information is encoded in the earlier layers, it is often lost or modified in the later layers (Kadavath et al., 2022; Azaria & Mitchell, 2023; Liu et al., 2023; Zou et al., 2023; CH-Wang et al., 2024). These studies primarily focus on linear probes of hidden representations (Ettinger et al., 2016; Conneau et al., 2018; Burns et al., 2023), but we demonstrate that the discrepancy between model output and internal structures can be captured through query-key interactions and attention maps.

## 3 ATTENTION AS SELECT-AND-COPY ALGORITHM

In this section, we describe how the attention mechanism can function as a *select-and-copy* operation. Consider a sequence of $N$ token embeddings, $\{\boldsymbol{x}_i\}_{i=1}^N$, which serve as the input to a given attention head in the transformer model, where each $\boldsymbol{x}_i \in \mathbb{R}^{d \times 1}$. In the classical transformer architecture Vaswani et al. (2017), each attention head performs a transformation of the input embeddings:

$$\boldsymbol{o}_m = \sum_{n=1}^N a_{m,n} \boldsymbol{v}_n, \quad a_{m,n} = \frac{\exp\left(\frac{\boldsymbol{q}_m^\top \boldsymbol{k}_n}{\sqrt{d}}\right)}{\sum_{j=1}^N \exp\left(\frac{\boldsymbol{q}_m^\top \boldsymbol{k}_j}{\sqrt{d}}\right)}, \tag{1}$$

where $\boldsymbol{q}_i = \boldsymbol{W}_q \boldsymbol{x}_i$, $\boldsymbol{k}_i = \boldsymbol{W}_k \boldsymbol{x}_i$, $\boldsymbol{v}_i = \boldsymbol{W}_v \boldsymbol{x}_i$ and $\boldsymbol{W}_q, \boldsymbol{W}_k, \boldsymbol{W}_v \in \mathbb{R}^{d_{model} \times d}$ are learned weight matrices. The resulting matrix $\boldsymbol{A} = \{a_{n,m}\}_{n,m=1}^N$ is stochastic, meaning that all its rows sum up to one. For decoder transformers, causal mask is applied to $\boldsymbol{A}$ before softmax: $a_{i,j} = 0, j > i$. Thus, from equation 1, for each token position $k$ in decoder transformers we can write

$$\boldsymbol{o}_m = \sum_{n \le m} a_{m,n} \boldsymbol{v}_n. \tag{2}$$

This means that the $m$-th token of the output embedding is a linear combination of the values of the preceding tokens, weighted by the $m$-th row of the attention matrix $\boldsymbol{A}$. If all but one component of this combination are close to zero, the transformation can be interpreted as a conditional copy mechanism. Specifically, if $a_{m,j}$ is the only non-zero weight in the $m$-th row, then $a_{m,j} \approx 1$, and $\boldsymbol{o}_m \approx \boldsymbol{v}_j$ (it follows from Eq. 2). Each token position from 0 to $m$ can be viewed as a cell storing the corresponding value vector, with the attention weights $a_{m,0}, \ldots, a_{m,m}$ determining the *choice* of which cell to copy to the $m$-th output.

Building on this, we propose the concept of *select-and-copy* heads, which implement this copying mechanism. Specifically, in this work, we aim to explore and identify attention heads within the

model that select the appropriate option and copy the corresponding information to the output. For such heads, the attention in the $m$-th row should be focused on a small set of selected tokens, where $m$ denotes the position of the output answer.

In modern models, positional encoding information is often incorporated as an additional transformation of queries and keys in Eq. 1. For example, in Rotary Position Embedding (RoPE) (Su et al., 2024), the rotation function $R_f(\cdot)$ is applied to queries and keys before the dot product is computed. The pre-softmax logit of standard attention becomes $R_f(\boldsymbol{q}_m)^T R_f(\boldsymbol{k}_n) = R_g(\boldsymbol{q}_m, \boldsymbol{k}_n, m - n)$, introducing a dependency on the position shift $m - n$.

In this paper, we evaluate the effectiveness of the answer option scoring derived from *select-and-copy* heads. Our goal is to identify heads that rely on the semantics of the options, rather than their positions. To mitigate the potential impact of the relative position shift, we introduce the *QK-score*, which does not incorporate positional shifts when comparing queries and keys (see details in the next section). We compare this score to the *Attention-score*, which is computed after RoPE is applied.

## 4 APPROACH

Consider an MCQA task with the corresponding dataset $\mathcal{D} = \mathcal{D}_{val} \cup \mathcal{D}_{test}$, where each instance consists of a prompt, a question, and labeled answer options (Fig. 1). Given this input, the model is tasked with generating the label for the best answer option.

To identify the heads in the model that implement the option selection mechanism described above, we first select the best-performing heads using $\mathcal{D}_{val}$, based on their accuracy on this validation set. We then evaluate their performance on the much larger $\mathcal{D}_{test}$. If these heads are indeed responsible for option selection, their performance should be at least comparable to that of the entire model. We confirm this claim through experiments presented in Section 5.4. An alternative method for selecting such heads is proposed in Section 6, where we demonstrate, through analysis of attention maps, that the best-performing heads effectively implement the option selection algorithm described above.

**QK-score and Attention-Score.** Given a data sample for an MCQA task, we denote by $q$ the question, which may be supported by context if applicable, by $o = \{o_1, o_2, ..., o_n\}$ the semantic content of the provided answer options, and by $d = \{d_1, d_2, ..., d_n\}$ the corresponding labels (e.g., A/B/C/D). We assume that the labels are ordered by default. These components are concatenated into a string: $q * d_1 * o_1 * \cdots * d_n * o_n *$, where $*$ represents any delimiter, typically punctuation marks or newline characters (Fig. 1). The model is tasked with estimating $P(d_i \mid q, d, o)$ – the probability of selecting option $d_i$ given the question $q$, the answer contents $o$, and the concatenated answer options $d$.

Let $t_i$, where $i \in \{1, 2, \ldots, n\}$, represent the indices of tokens that encode information about the corresponding answer options. We refer to these as *option-representative tokens*. Properly selecting such tokens is crucial for the success of our algorithm. In most experiments, we use the end-of-line token following the $i$-th option content as $t_i$; alternative choices are presented in Fig.2a, and we analyze them in Sec.6.

Let $N$ denote the length of the entire text sequence, and consider the attention head with index $h$ from layer $l$. Given the triplet $(q, d, o)$, we can compute the *QK-score* $S_{QK}^{(l,h)}(d_i)$ for option $d_i$ based on the query and key vectors, as well as the *Attention-score* $S_{Att}^{(l,h)}(d_i)$ derived from the attention weights:

$$S_{QK}^{(l,h)}(d_i) = \boldsymbol{q}_N^{(l,h)\top} \boldsymbol{k}_{t_i}^{(l,h)}, \quad S_{Att}^{(l,h)}(d_i) = a_{N,t_i}^{(l,h)}, \quad i \in \{1, 2, ..., n\} \tag{3}$$

The *QK-score* for the $i$-th option is calculated as the dot product of the $t_i$-th key and the final query vector $q_N$ (see Fig. 1). In the *QK-score*, we do not apply a positional transformation; therefore, it does not correspond to the attention scores prior to the softmax operation. As a result, the token with the highest *QK-score* does not necessarily correspond to the token with the maximum attention score. For an example, see Figure 7.

For each method, the prediction is straightforward: we select the option with the highest score. Additionally, by applying the softmax function to the scores, we can estimate the probabilities for each option, specific to the attention head.

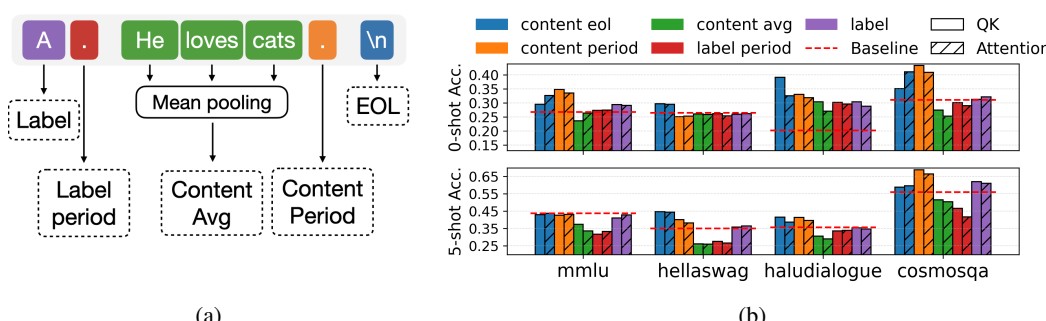

(a)                 (b)

Figure 2: (a) Scheme for option-representative token types. (b) Performance of *QK-score* and *Attention-score* for different option-representative tokens on Llama2-7B base.

**Choosing the predicting heads.** For predictions we use the scores from the single best head, which is selected based on accuracy on the validation set $\mathcal{D}_{val}$. In Section 5.4, we report the results obtained from the best head chosen separately for each dataset and each number of shots (i.e., the number of examples provided in the prompt). In Section 6, we demonstrate that for each model, there exist universal heads that perform well across most tasks and numbers of shots. Furthermore, we show that these universal heads can be identified without access to labeled validation data. We do not aggregate predictions from multiple heads; instead, we consider this as a direction for future research.

## 5 EXPERIMENTS

### 5.1 DATASETS

We conduct experiments on four challenging real-world MCQA datasets from LLM benchmarks: **MMLU** (Hendrycks et al., 2021), **CosmosQA** (Huang et al., 2019), **HellaSwag** (Zellers et al., 2019) and **HaluDialogue**, which is a "dialogue" part of HaluEval (Li et al., 2023a). All of them consist of questions with four possible answer options, and some also include a context that must be used to determine the correct answer. More details about each dataset can be found in Appendix A.1. Additionally, we introduce **Simple Synthetic Dataset** (SSD), a synthetic task in the MCQA setting designed to evaluate the model's ability to handle the basic task format. Tasks in the SSD do not require any factual knowledge from the model. The main version of this dataset consists of questions in the form, "Which of the following options corresponds to "<word> ?" and contains 2.500 examples. The options consist of a word from the question and three random words, all mixed in a random order and labeled with the letters 'A'-'D'. Other variations of this dataset contain smaller versions in different languages (see Appendix K), as well as versions with a different number of options and different data labels, all sampled and named according to the same principle (see Appendix G for more details).

Finally, following Ye et al. (2024) in all five datasets we specially modified questions by adding two extra options "E. None of the above." and "F. I don't know." that are intended to aggregate the uncertainty of LLM. Despite adding these two options, there are *NO* questions for which 'E' or 'F' are correct answers. Examples and prompts are listed in the Appendix A.2.

Following the previous approach by Zheng et al. (2024a), with fixed $N$-shot setup, we select $\mathcal{D}_{val}$ as 5% of $\mathcal{D}$ for each dataset that is dedicated to assessing each head's performance. Based on this evaluation, the best head is chosen and applied to other questions in the dataset.

### 5.2 BASELINES

The standard approach for MCQA is to use output probabilities from LLM for all options $d_i$ to choose the predicted option $\hat{d}$:

$$\hat{d} = \arg\max_{d_i} P(d_i \mid q, d, o), \tag{4}$$

where $q$ is the question, $o = \{o_1, o_2, ..., o_n\}$ are the option contents, and $d = \{d_1, d_2, ..., d_n\}$ are the options labels (e.g A/B/C/D). In our experiments, we refer to this method as `Baseline`.

In recent work (Zheng et al., 2024a), it was proposed to mitigate option selection bias by averaging the results over option permutations. The idea is to use the set of all cyclic permutations $\mathcal{I} = \{(i, i+1, ..., n, 1, ..., i-1)\}_{i=1}^n$ to calculate the debiased probability:

$$\tilde{P}(d_i \mid q, d, o) = \frac{1}{|\mathcal{I}|} \sum_{I \in \mathcal{I}} \log P(\pi_I(d_i) \mid q, d, \pi_I(o)) \tag{5}$$

Since computing probabilities for all permutations for each question is expensive, authors propose to estimate the prior distribution for option IDs on test set which contains $5\%$ of all samples, and use it to debias new samples. In our experiments we refer to this method as `PriDe`. The test set is the same as the one that we use for the best heads selection. We employ this method to evaluate the QK-score's effectiveness in addressing selection bias and to compare it with the model's performance after debiasing. While these methods could potentially be combined to enhance results, exploring such integrations is left for future research.

## 5.3 EXPERIMENTAL SETUP

Our main experiments were carried out according to the following pipeline: first, we took a frozen pre-tranied Transformer LLM (its weights were not modified in any of the experiments). We then passed questions from the validation subset through the model and for each attention head and each question, we obtained the best answer in terms of *QK-score*. After that, we selected a single head on which the highest accuracy was achieved. If several heads had equal accuracy scores, we chose one from the lower level of the model (although this occurred extremely rare in our experiments). Next, we obtained the predictions using both the baseline method and the *QK-score* on the chosen head. Finally, we perform random shuffle of options in all questions and repeat the above procedure; this was done to correctly compute the Permutation Accuracy metric. Note that it may be two different heads that achieve best *QK-scores* on validation set before and after the option permutation.

We report two quality metrics on the test subset: the accuracy of predicted answers (from the first run) and the *Permutation Accuracy* (PA). The latter was introduced in Gupta et al. (2024) and is, in a sense, accuracy stable with respect to option permutation. PA metric is computed as the percentage of questions for which the model selects the correct choice both before and after the random permutation of options. $\text{PA} = \frac{1}{N} \sum_{i=1}^N \text{I}_i \text{I}_i^p$, where $N$ is the dataset size, $\text{I}_i$ is the indicator value equals to 1 iff the model answers question $i$ correctly, while $\text{I}_i^p$ equals to 1 iff the model answers question $i$ correctly after its options (not the letters, but their texts) were permuted. At the same time, answer options "E. None of the above." and "F. I don't know." are special and therefore are exempt from shuffling.

The prompt templates used in our experiments are provided in Appendix A.3. In the few-shot regimes, before asking the question we provide the model with demonstrations in the same format, except that the true answers (a single capital letter for the correct option) are given after each example, separated by a single whitespace. The examples are separated from each other and from the actual question by single line breaks. The demonstrations are the same for every question in the given dataset. The set of examples for $(k+1)$-shot prompts contains the set of examples for the $k$-shot prompts, plus one new example. The demonstrations were chosen from the first fifteen entries of the validation set. Their selection was mostly arbitrary, but we tried to filter out questions that we considered suboptimal from the perspective of an English-speaking human expert.

## 5.4 RESULTS

Figure 3 demonstrates the results of our method for LLaMA2-7B model. We observe an impressive improvement by 7-16% on all the datasets in the zero-shot regime. Although *QK-scores* are not completely robust to option permutations, they are more stable than the baseline: the relative performance drop, measured by the PA metric, is smaller than for the baseline across all datasets. In the few-shot regime, our approach is on par or outperforms other methods, with the most visible improvement on Halu Dialogue dataset by 5-9% depending on the number of shots.

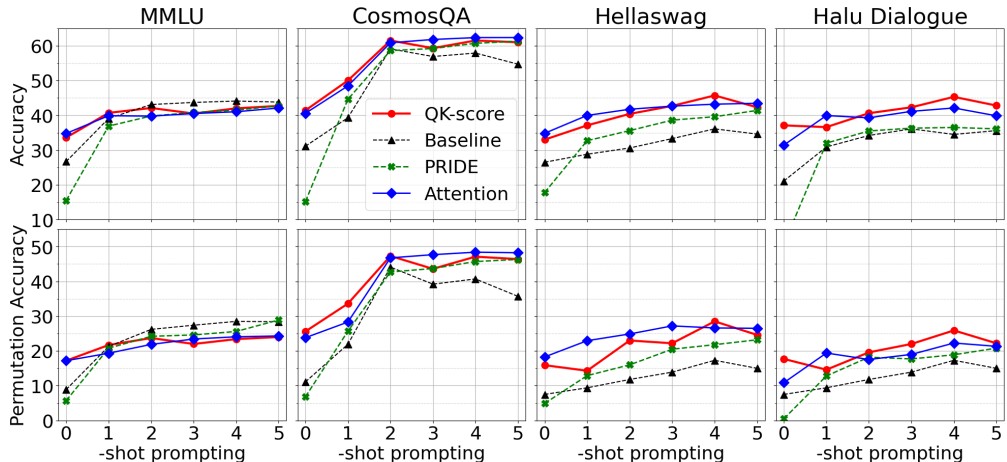

Figure 3: Comparison of different methods for LLaMA2-7B (base) on various Q&A datasets. Reported metrics are Accuracy (Acc) and Permutation Accuracy (PA).

PriDe results are added in Figure 3 for comparison. In most cases, PriDe performs better than the baseline, but sometimes fails in the zero-shot regime. Our analysis reveals that this method is not robust for the additional uncertain options "E" and "F". Additionally, we provide experiments without such options in Appendix B, where PriDe performs better in the few-shot regimes, but still loses in the zero-shot. But overall, in all cases and for any option sets, QK score outperforms PriDe.

We also applied our method to larger models of LLaMA family: LLaMA2 (-13B, -70B) and LLaMA3 (-8B, -70B) as well as to their chat/instruction-tuned versions. Table 1 presents the results of our method for large models in zero-shot regime; full version including the few-shot regimes is provided in Appendix H. Aside from this, we conducted experiments on the models from Qwen 2.5 family with number of parameters ranging from 1.5B to 14B (Team (2024b)), as well as on three small models from other families (ranging from 2B to 3.8B): Gemma 2-2B (Team (2024a)), Dolly V2-3B (Conover et al. (2023)), and Phi-3.5-mini (et. al. (2024)). These results can be found in Appendix L . Overall, the results are mostly in line with those obtained for LLaMA2. For all relatively smaller models of 8B and 13B size in the zero-shot settings, our approach outperforms the baseline on all datasets, both in terms of accuracy and permutation accuracy, with the improvement up to huge 27%, achieved on HellaSwag dataset with LLaMA3-8B model. This effect is not as clearly expressed for models of very small size, i.e. Qwen 2.5-1.5B; see Appendix L for discussion. With larger models, MMLU is the most difficult benchmark for our method, likely because its questions are aimed at measuring general knowledge, while our method, by design, focuses more on the semantic relations between the question and the possible answers.

Regarding the performance of models on the synthetic dataset, Figure 5b shows that, in the baseline zero-shot setting, LLaMA2-7B struggles to select the correct option. In contrast, our method enables the extraction of the necessary information from the model, thus resulting in much better performance. The figure displays accuracy for the *QK-score* from the five best heads (denoted by their *(Layer, Head)* indices). Three of these heads are also shown in Table 9, while the other two—(8, 8) and (12, 15)—are unique to this particular dataset.

## 6 ANALYSIS

**Choosing option-representative tokens.** To compare *QK- and attention* scores, we need to select option-representative tokens $\{t_i\}$, where the semantic information about each option semantics is concentrated. Due to the causal nature of attention in LLMs, the logical choice is the last token after the content of the option, which is the end-of-line token. We use it in most of our experiments, although there are other tokens worth analysing: the label itself, the period after the label, and the period after option content (see Fig. 2a). We also experimented with the mean aggregated score for all tokens in the option's content, but this gave poor results. The detailed analysis of such variations

| Method | | -30B | -65B | 2-13B | LLaMA...
2-70B | 3-8B | 3-70B | 2-13B | LLaMA... (chat, instruct)
2-70B | 3-8B | 3-70B |
|---|---|---|---|---|---|---|---|---|---|---|---|
| | | | | | **MMLU** | | | | | | |
| Baseline | Acc | **50.4** | **48.3** | 34.6 | **59.7** | 60.3 | **75.3** | 47.4 | 57.7 | 60.5 | **78.2** |
| | PA | **37.9** | **35.7** | 22.4 | **48.5** | 50.4 | **68.8** | 34.6 | 45.9 | 47.7 | **70.1** |
| QK-score | Acc | 45.2 | 46.2 | **42.2** | 56.7 | **61.0** | 74.5 | **49.7** | **58.9** | **63.0** | 77.9 |
| | PA | 30.7 | 32.1 | **25.9** | 39.2 | **51.5** | 66.0 | **38.3** | **47.1** | **49.3** | 67.9 |
| | | | | | **Cosmos QA** | | | | | | |
| Baseline | Acc | 59.9 | **65.7** | 29.6 | 65.5 | 54.9 | 82.0 | 48.1 | 68.5 | 85.4 | 91.6 |
| | PA | 47.5 | **53.1** | 19.4 | **56.3** | 39.3 | 75.7 | 36.8 | 58.3 | 71.0 | 82.5 |
| QK-score | Acc | **60.1** | 63.5 | **58.2** | **69.5** | **70.6** | **87.6** | **67.7** | **84.8** | **88.6** | **94.1** |
| | PA | 44.4 | 50.8 | **44.3** | 56.2 | **60.9** | **81.7** | **51.6** | **75.9** | **75.1** | **88.1** |
| | | | | | **Hellaswag QA** | | | | | | |
| Baseline | Acc | 35.2 | 33.4 | 36.8 | 71.6 | 33.5 | **82.5** | 41.6 | 61.4 | 67.4 | **86.8** |
| | PA | 16.5 | 13.7 | 17.1 | 62.9 | 15.8 | **76.1** | 25.8 | 49.0 | 27.8 | 71.2 |
| QK-score | Acc | **43.9** | **53.8** | **52.9** | **74.9** | **60.9** | 82.1 | **50.8** | **73.0** | **72.5** | 86.3 |
| | PA | **21.5** | **35.0** | **38.8** | **63.3** | **50.8** | 75.2 | **37.3** | **64.9** | **36.3** | **72.8** |
| | | | | | **Halu Dialogue** | | | | | | |
| Baseline | Acc | 36.3 | **46.7** | 41.0 | 39.4 | 46.6 | 44.3 | 49.4 | 39.4 | 62.1 | 68.8 |
| | PA | 21.1 | **29.8** | 22.2 | 25.4 | 29.1 | 33.5 | 32.6 | 26.6 | 42.6 | 63.8 |
| QK-score | Acc | **44.8** | 42.4 | **47.2** | **58.4** | **52.3** | **67.8** | **56.2** | **58.1** | **64.7** | **76.7** |
| | PA | **27.6** | 22.5 | **30.2** | **42.6** | **36.7** | **57.9** | **42.5** | **42.8** | **46.6** | **65.6** |

Table 1: Comparison of different base models in zero-shot setup on various Q&A datasets. Reported metrics are Accuracy (Acc) and Permutation Accuracy (PA). Best results are highlighted in **bold**.

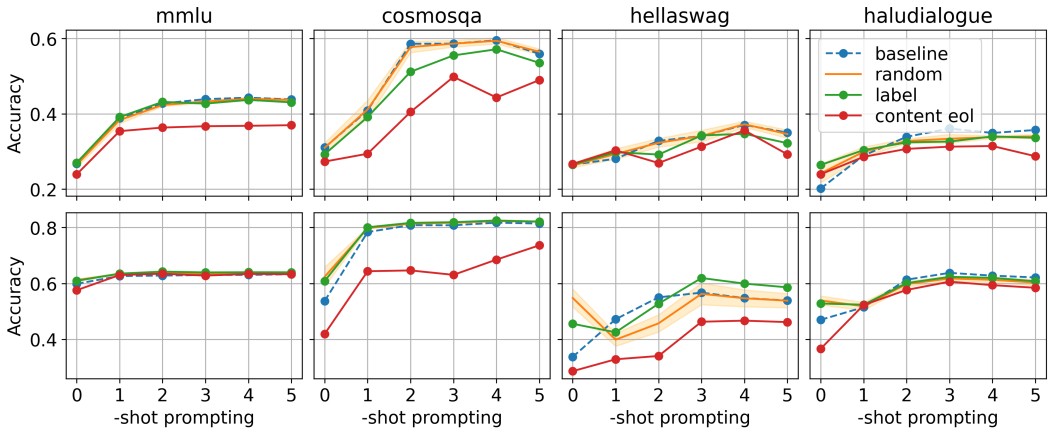

Figure 4: Zero-ablation of heads for LLaMA2-7B (upper) and LLaMA3-8B (lower)

for attention scores is presented in Fig. 2b. We observe that the period after content and the end-of-line tokens are the most representative for our scores. There is an interesting finding concerning the label token: despite it being almost useless in the 0-shot setup, what is in line with (Lieberum et al., 2023), it shows good performance for the 5-shot setup on different heads. We hypothesise that there are several types of "select-and-copy" heads, which influence the logits differently.

**Select-and-copy heads ablation.** We investigate the relationship between *select-and-copy* heads and model performance using zero-ablation of heads (Olsson et al., 2022), where we replace the output of selected heads with a zero vector to isolate their effect on the model's output. We focus on the 10 best-performing heads, based on the *Attention-score*, and report the results in Fig. 4. Additional experiments, including logit lens analysis (Nostalgebraist, 2020), are provided in Appendix F.

**Best heads.** A validation set of substantial size is required to choose the best head; therefore, we would like to determine whether universal heads exist, that perform on par with those calibrated for

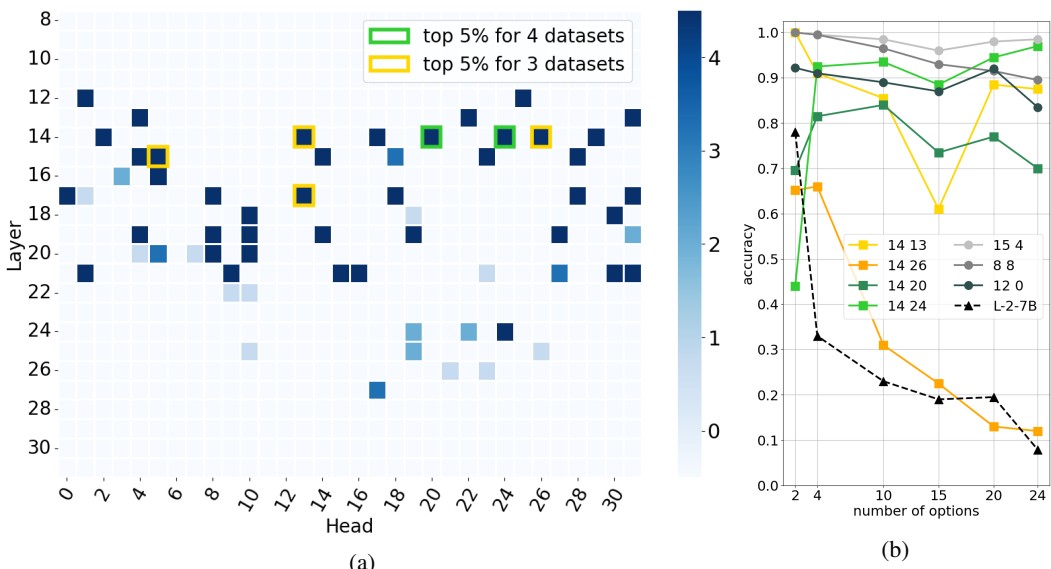

(a)  (b)

Figure 5: (a) Heatmap showing the (layer, head) indices of the best-performing heads in LLaMA2-7B. The colored frames highlight heads that generalize well across different datasets, with the color intensity indicating the level of generalization across varying numbers of demonstrations. (b) QK-score accuracy for a synthetic dataset with varying numbers of options (2 to 24) in a zero-shot setting for LLaMA2-7B. The x-axis represents the number of options, and the y-axis represents accuracy. Colored lines represent different attention heads. "Square" markers indicate heads that perform well on real datasets (highlighted in Figure 5a); "round" markers highlight heads that perform well specifically on the synthetic dataset; "triangle"-dotted line shows the baseline performance.

specific topics. Identification of such heads would also help mitigate the impact of poor choice of the validation set, when there are discrepancies between the questions in it and in the test set.

To illustrate how the best-performing heads vary between different setups, we study separately the heads generalization across different datasets, and across different number of demonstration examples. We select the top 5% heads best in terms of the mean accuracy for each mix of datasets and each combination of in-context examples counts (see Appendix D for more details). The results are shown in Figure 5a. This heatmap highlights the most stable heads, which appear among the best in several mixed tasks: when "shots" are mixed (framed cells), or when datasets are mixed (coloured cells). Notably, the majority of the robust heads in this sense lay between layers 12 and 21. The most universal heads w.r.t. to dataset change are (14,24) and (14,20). They appeared in the top 5% pairs in mixed-shot setup for all four datasets. They also demonstrate high performance on the synthetic data when the number of options is increased to 24, as shown in Figure 5b; at the same time, the performance of the baseline method drops below that of random choice.

Moreover, these heads appear among the top 10 performing heads even for versions of SSD in other languages, such as Italian, French, and Russian (see Appendix K). These results provide additional evidence that the selected heads are indeed capable of performing the option selection task based on option content. More detailed analysis of 0-shot performance is provided in Appendix E.

**Attention patterns analysis.** Figure 6 shows the typical attention pattern together with *QK-scores* for our most stable head (14, 24); attention patterns of other top-performing heads — (14, 20), (14, 26), and (14, 13) (top right corner of the Figure 10) — are shown in Appendix, Figure 7. We can see that the attention weights are concentrated on option-representative tokens, namely $\backslash n$ symbols after the options, with the highest weight on the correct one. This is exactly what is be expected from *select-and-copy* heads. Interestingly, the *QK-scores* provide a clearer picture of this phenomenon.

**Finding best heads without validation labels.** Based on this observation, we propose an algorithm to identify such stable heads without the need for a labeled validation set. Specifically, such heads

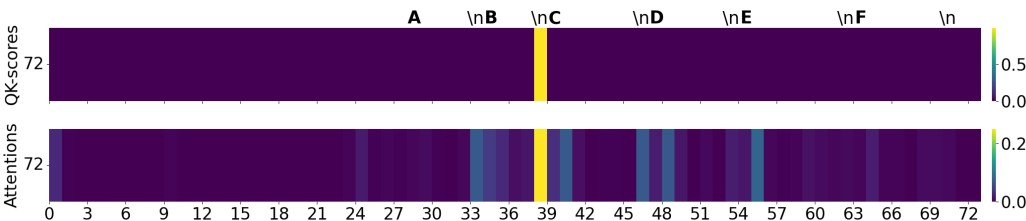

Figure 6: QK-scores after softmax (upper part of the diagram) and attentions (lower part of the diagram) for the last token on the 0-shot MMLU example on (14, 24) head. The task is "Question: What singer appeared in the 1992 baseball film 'A League of Their Own'\nOptions:\nA. Brandy.\nB. Madonna.\nC. Garth Brooks.\nD. Whitney Houston.\nE. I don't know.\nF. None of the above.\nAnswer:". Full version is in Figure 7 in the Appendix.

should have high sum of attention weights on option-representative tokens and high variability in the options they focus at (see formal definition in Appendix I). When we sort all the heads of LLaMA2-7B model by this score, we find that the four previously identified heads are ranked very high. The heads (14,20) and (14,24) are especially stable across datasets, number of shots and options (Fig. 5a).

**Selection bias.** Following previous studies on selection bias Pezeshkpour & Hruschka (2024); Zheng et al. (2024a), we investigate our methods in relation to the tendency to choose specific option rather than the correct answer. We observe that, among the best heads, there is also uneven distribution in predictions, which is mitigated when the number of in-context examples is increased. However, an interesting pattern emerges: the distributions of two best heads complement each other, e.g. $S_{QK}^{(14,20)}$ is biased to options "A" and "D" an $S_{QK}^{(14,24)}$ — to options "B" and "C". More detailed information provided in Figure 40 in Appendix J .

## 7 CONCLUSION

In this work, we introduced two novel scoring mechanisms: *QK-score* and *Attention-score*, derived from internal mechanism of LLM that can help to improve the performance on multiple-choice question answering tasks. Our experiments demonstrated significant improvements (up to 16%) across popular benchmarks, and even more striking results (up to 60%) on a synthetic dataset designed to test the model's understanding of task format.

We identified a subset of attention heads, which we termed *select-and-copy* heads that play a critical role in these performance gains. These heads are relatively stable across different datasets and exist universally across model scales, and we explored their causal effect on task performance. Our findings suggest that these specialized heads have the potential to deepen our understanding of LLMs' capabilities not only for MCQA but for other reasoning tasks as well.

This work opens up new avenues for further research into the internal dynamics of LLMs, including a deeper exploration of attention mechanisms and their role in complex task-solving that requires selection and copying information from the text.

## 8 LIMITATIONS

Our method cannot be applied to models without an access to attention matrices. Also, our method is not applicable on scarce-resource tasks, even though one can utilize the heads we marked as robust enough. Besides, MCQA task itself was criticized for oversimplification (Balepur et al., 2024).

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

## A  DATASETS

### A.1  DATASETS DETAILS

**Massive Multitask Language Understanding (MMLU)** (Hendrycks et al., 2021) contains 4-way questions on the variety of topics related to STEM, the humanities, the social sciences, and other fields of knowledge. We sample 10,000 instances from the test set to utilize them in our experiments.

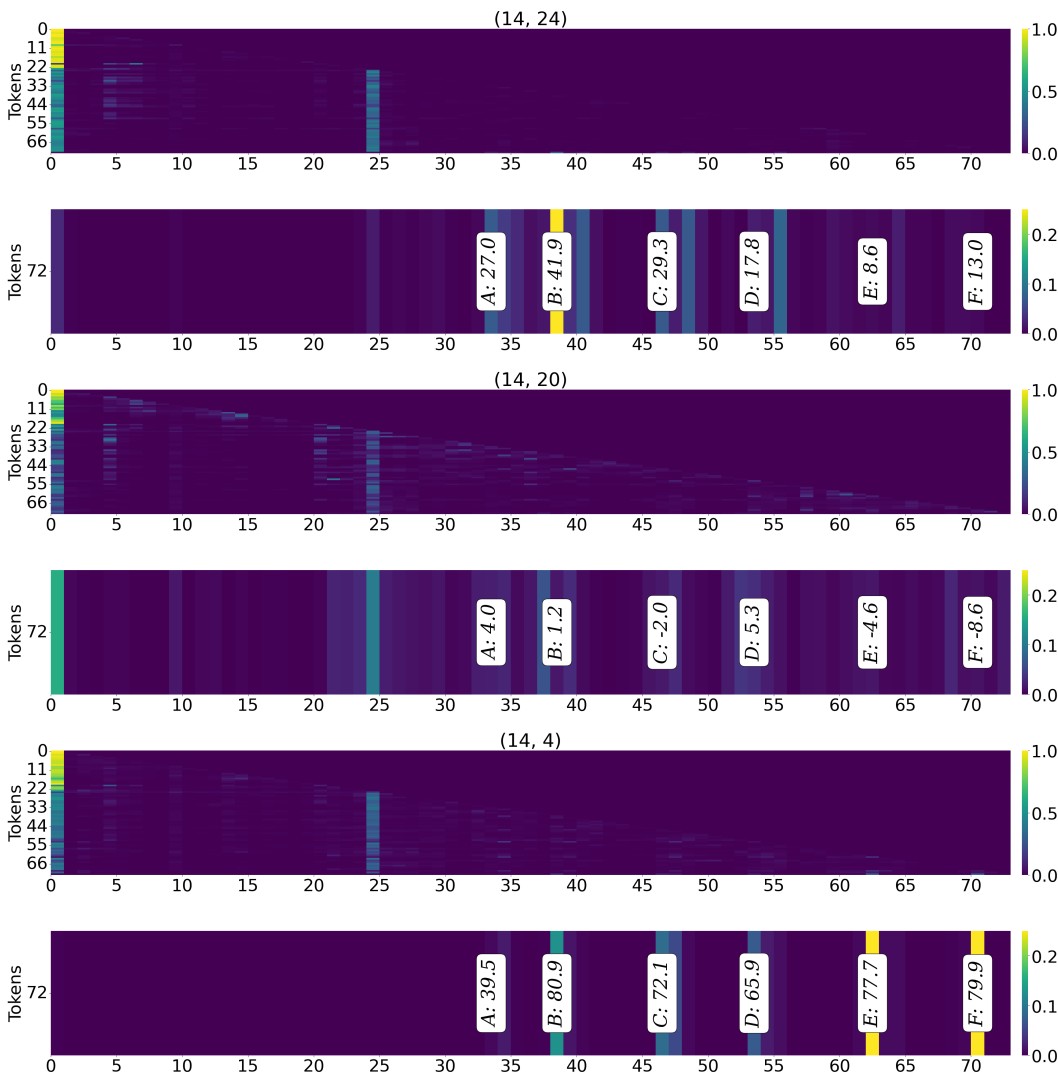

Figure 7: Attention maps of (14, 24), (14, 20) and (14,4) pairs (Head, Layer) for 0-shot setting for MMLU example: `Question: What singer appeared in the 1992 baseball film 'A League of Their Own'? \nOptions: \nA. Brandy.\nB. Madonna.\nC. Garth Brooks.\nD. Whitney Houston.\nE. I don't know.\nF. None of the above.\nAnswer:.` Second plot for each pair corresponds to the same, but scaled to the end-of-text-sequence attention map. Values in annotated cells are corresponding QK-score values. End of each option is denoted with \n symbols. 33th token is the end of A option, 38th token is the end of B option, 46th token - the end of C option, 53th token - the end of D option, 62th token - the end of E option, 70th token - the end of F option. The answer from QK-score of (14, 24) and (14, 4) is B, of (14, 20) is D. The correct answer for this example is B.

**CosmosQA**[1] (Huang et al., 2019) together with question and answer options additionally contains text paragraph that is supposed to be used by a model to give the final answer. The purpose is to evaluate the model's reading comprehension and commonsense reasoning capabilities. Similar to MMLU, we sampled 10,000 instances from the test set.

---

[1]https://wilburone.github.io/cosmos/

**HellaSwag** (Zellers et al., 2019) evaluates the commonsense reasoning capabilities of the model by selecting the best sentence completion for a given sentence prompt, given a short text as a context. We also extracted 10,000 entities from this dataset.

**HaluDialogue** is a "dialogue" part of HaluEval (Li et al., 2023a) dataset with about 10,000 examples. Here, a model is asked to choose an appropriate continuation of a dialogue from four possible options.

We chose datasets in order to cover the main formats of questions and common NLP tasks. Since our primary intention was to focus on the investigation and interpretability of attention heads' roles in Question Answering, we limited ourselves to these four datasets. We did not try to cover as many benchmarks as possible.

## A.2 EXAMPLES OF QUESTIONS FROM DATASETS

```
Question: Where is the Louvre museum?
Options:
    A. Paris.
    B. Lyon.
    C. Geneva.
    D. Vichy.
    E. I don't know.
    F. None of the above.
```

Listing 1: MMLU example

```
Context: My house is constantly getting messy and I ca n't keep up . I am
    starting at a new school with no one I know and it is 4 times bigger
    than UAF . I am now going to have to balance school , homework ,
    kids , bill paying , appointment making and cleaning when I can
    barely keep up without the school and homework ( keep in mind this is
     a full time GRADUATE program at a fairly prestigious school ) . We
    are in financial crisis .
Question: What is causing the narrator 's recent stress ?
Options:
    A. They are moving to a new house .
    B. I would have tried to guess their password and alternatively gone
    to a coffee shop for wifi.
    C. They are moving to a new university .
    D. They are moving to a new house for the kids .
    E. I don't know.
    F. None of the above.
```

Listing 2: CosmosQA example

```
Context: A young boy is wearing a bandana and mowing a large yard. he
Question: Which of the following is the best ending to the given context?
Options:
    A. is unrelieved by the weeds and is barely smiling.
    B. walks away from the camera as he pushes the mower.
    C. moves and walks the mower but gets stuck because he is engaged in
    a game of ping pong with another boy.
    D. seems to be doing a whole lot of things and talks to the camera
    from behind a white fence.
    E. I don't know.
    F. None of the above.
```

Listing 3: HellaSwag example

```
Context: [Human]: I like Pulp Fiction. What do you think about it? [
    Assistant]: I love it. It was written by  Roger Avary [Human]: I
    heard he also wrote The Rules of Attraction. Do you know who is in
    that movie?
```

```
Question: Which of the following responses is the most suitable one for
    the given dialogue?
Options:
    A. Swoosie Kurtz is in it.
    B.  Fred Savage is in it.
    C. Yes, it is a drama and crime fiction as well. Do you like crime
    fiction stories too?.
    D. No, it was not made into a film. However, it was adapted into a
    popular Broadway musical.
    E. I don't know.
    F. None of the above.
```

Listing 4: Halu Dialogue example

```
Question: Which of the following options corresponds to " optimal "?
Options:
    A. ion.
    B. optimal.
    C. coins.
    D. jackie.
    E. I don't know.
    F. None of the above.
```

Listing 5: Simple Synthetic Dataset example

### A.3 PROMPT TEMPLATES AND EXAMPLES

Variable parts are highlighted in **bold**; whitespace placing is marked by underscores; the position of line breaks is explicitly shown by symbols '\n' (note that the last line always ends without whitespace or line break). In our datasets, we ensured that each question ends with a question mark, and each choice ends with a point (a single whitespace before it does not affect the logic of tokenization by the LLaMA tokenizer).

```
Question:␣{Text of the question}?\n
Options:\n
A.␣{Text of the option A}␣.\n
B.␣{Text of the option B}␣.\n
C.␣{Text of the option C}␣.\n
D.␣{Text of the option D}␣.\n
E.␣I␣don't␣know␣.\n
F.␣None␣of␣the␣above␣.\n
Answer:
```

Listing 6: MMLU prompt template

```
Context:␣{The context of the question/situation or the dialog history}\n
Question:␣{Text of the question}?\n
Options:\n
A.␣{Text of the option A}␣.\n
B.␣{Text of the option B}␣.\n
C.␣{Text of the option C}␣.\n
D.␣{Text of the option D}␣.\n
E.␣I␣don't␣know␣.\n
F.␣None␣of␣the␣above␣.\n
Answer:
```

Listing 7: CosmosQA/HellaSwag/Halu Dialogue prompt template

The following is an example of a 1 shot prompt from MMLU. 2-3-4-5-shot prompts were built in the same way, and prompts for datasets with context were built the same way, except each question is preceded by its context. Note that in demonstrations, we add a single whitespace between "Answer:" and the correct choice letter; for example, "Answer: A", but *NEVER*

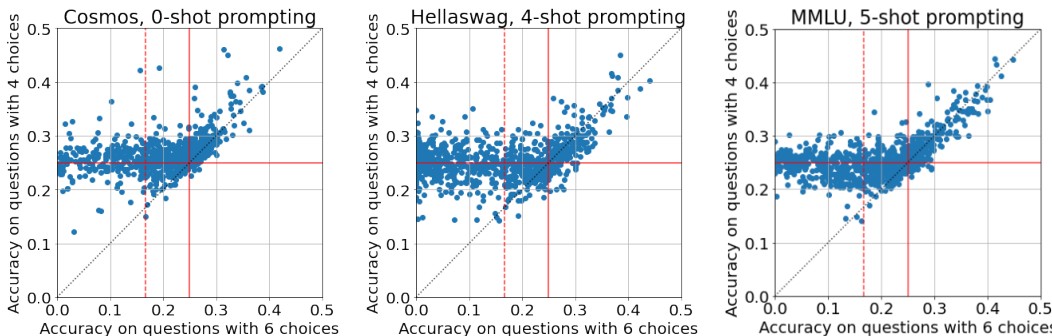

Figure 8: Correlation between heads QK-scoring accuracy on questions with 4 ('A'-'D') and 6 ('A'-'F') answer options. Solid red lines mark the accuracy level of 0.25, dashed red line – 0.167 (6 options random choice accuracy).

"`Answer:A`". This is done because sequences like "`:  A`" and "`:A`" are differently split into tokens by the LLaMA tokenizer. The former produces the same tokens corresponding to the letter "`A`" as in the choice option line, while later yields a different version of "`A`". From LLaMA's point of view, these two versions of letters are separate entities and are NOT interchangeable. Removing those symbols of whitespace often leads to a noticeable drop in performance.

```
Question:␣A␣medication␣prescribed␣by␣a␣psychiatrist␣for␣major␣depressive␣
    disorder␣would␣most␣likely␣influence␣the␣balance␣of␣which␣of␣the␣
    following␣neurotransmitters?\n
Options:\n
A.␣serotonin␣.\n
B.␣dopamine␣.\n
C.␣acetylcholine␣.\n
D.␣thorazine␣.\n
E.␣I␣don't␣know␣.\n
F.␣None␣of␣the␣above␣.\n
Answer:␣A\n
Question:␣
    Meat should be kept frozen at what temperature in degrees Fahrenheit?
    \n
Options:\n
A.␣0 degrees or below␣.\n
B.␣between 10 and 20 degrees␣.\n
C.␣between 20 and 30 degrees␣.\n
D.␣0 degrees or below␣.\n
E.␣I␣don't␣know␣.\n
F.␣None␣of␣the␣above␣.\n
Answer:
```

Listing 8: An example of 1-shot prompt for a question from MMLU dataset

## B  SOME MORE INTUITION ON OPTIONS 'E' AND 'F'

As mentioned in the main text, including fictional, though always incorrect, choices "`E. None of the above`" and "`F. I don't know`" in every question was aimed at creating the "uncertainty sinks". However, they are also beneficial for analyzing attention head roles, but that is somewhat beyond the scope of this article. Here, we would like to provide some intuition about it.

We performed experiments on a modified version of our datasets, where questions include only 4 "meaningful" choices, i.e., options 'A'-'D' only. Scatterplots in Figure 8 show the correlation between the accuracy of heads using QK-scores on options without 'E'-'F' (by y-axis) and their accuracy on questions with all six options (by x-axis). Here, only validation subsets were used. We present plots for some possible setups, but others follow similar patterns. From these charts, we can

see that if a head reaches good accuracy answering 4-choice questions, it usually will reach nearly the same accuracy on questions with six choices and vice versa; see points around the diagonal $y = x$ in the upper-right quadrant.

We can also observe another significant trend: horizontal stripe near y-level $0.25$. It can be explained in the following manner: in the data used, ground-truth answers are perfectly balanced – that is, for every choice 'A'-'D' $25\%$ of the questions have it as the correct answer. Therefore, if a head reaches 4-choice accuracy of $\approx 25\%$, it falls into one of the three categories:

1. This head chooses only one option in all questions. Usually, it is the last one on the list.
2. This head "guesses" answers, choosing options nearly randomly and "independent" from their meanings.
3. This head "understands" questions but is genuinely bad at answering them.

The addition of choices 'E' and 'F' drops the performance of the first type heads down to nearly $0\%$, second type – to around $16.7\%$; QK-scoring accuracy of the third type heads, however, usually remains the same.

Thus, we can conclude that choices 'E' and 'F' cause little effect on the performance of good heads, but, at the same time, their inclusion creates separation between heads that are bad at Multiple Choice Question Answering and heads that do not have MCQA in their functionality at all (they may perform other roles for LM).

## C  NUMERICAL RESULTS FOR COMPARISON OF QK-SCORE WITH OTHER METHODS

Table 2 provides numerical results for our main experiments with QK-scores from heads of the LLaMA2-7B model that are presented in Figure 3 in the main text.

## D  BEST HEADS

| Setup | Best (Layer, Head) |
|-------|-------------------|
| 0-shot | **(14, 24)** |
| 1-shot | (15, 5), **(15, 23)**, (14, 20) |
| 2-shot | **(14, 24)**, (15, 5), (15, 4) (18, 10), **(15, 23)**, (16, 17) |
| 3-shot | **(14, 24)**, (15, 5), (15, 4) (18, 10), **(15, 23)**, (14, 26), (17, 18) |
| 4-shot | **(14, 24)**, (15, 5), (14, 4) (15, 4), (18, 10), **(15, 23)** (14, 20), (14, 26), (17, 18), (16, 17) |

(a) Mixed top heads based on datasets

| Dataset | Best (Layer, Head) |
|---------|-------------------|
| MMLU | **(14, 24)**, (15, 4), (17, 0), **(14, 20)**, (20, 10), (18, 30) |
| HaluDialogue | (14, 29), **(14, 24)**, (14, 26) |
| HellaSwag | (15, 5), (15, 4), (18, 10), **(14, 20)**, (14, 13), (13, 22) |
| CosmosQA | **(14, 24)**, (15, 5), (15, 4), (18, 10), (17, 0), (15, 23), **(14, 20)**, (14, 26), (14, 13), (18, 30) |

(b) Mixed top heads based on shots.

Figure 9: Top 1% heads based on accuracy for different ways of mixing

```
mmlu_top_heads = {
    0: [(14, 24), ...],
    1: [(14, 20), ...], ... # for all 5 shots  top 1% heads for MMLU
}

hellaswag_top_heads = {
    0: [(15, 10), ...],
    1: [(14, 20), ...], ... # for all 5 shots top 1% heads for HellaSwag
}

top_heads = [[] for i in range(5)]  # 0 shot to 4 shot
```

| Method | | ...-shot prompting | | | | | |
|---|---|---|---|---|---|---|---|
| | | 0 | 1 | 2 | 3 | 4 | 5 |
| **MMLU** | | | | | | | |
| Baseline | Acc | 26.7 | 39.1 | **43.1** | **43.7** | **44.1** | **43.8** |
| | PA | 8.9 | 21.3 | **26.2** | 27.4 | **28.5** | **28.4** |
| PRIDE | Acc | 15.5 | 36.9 | 39.8 | 40.8 | 41.5 | 42.7 |
| | PA | 5.7 | 20.8 | 24.2 | 24.6 | 25.6 | 28.9 |
| Attention | Acc | **34.8** | 39.9 | 39.8 | 40.5 | 41.0 | 42.1 |
| score | PA | **17.2** | 19.4 | 21.9 | 23.4 | 24.1 | 24.3 |
| QK-score | Acc | 33.6 | **40.7** | 42.1 | 40.5 | 42.0 | 42.7 |
| | PA | **17.2** | **21.7** | 23.7 | 22.0 | 23.4 | 24.0 |
| **Cosmos QA** | | | | | | | |
| Baseline | Acc | 31.1 | 39.3 | 59.1 | 56.9 | 57.9 | 54.7 |
| | PA | 11.1 | 21.9 | 44.1 | 39.2 | 40.7 | 35.7 |
| PRIDE | Acc | 15.2 | 44.6 | 58.6 | 59.2 | 60.7 | 61.3 |
| | PA | 6.8 | 25.7 | 42.7 | 43.7 | 45.7 | 46.3 |
| Attention | Acc | 40.6 | 48.5 | 60.9 | **61.8** | **62.3** | **62.3** |
| score | PA | 23.8 | 28.3 | 46.8 | **47.7** | **48.4** | **48.2** |
| QK-score | Acc | **41.4** | **50.0** | **61.5** | 59.3 | 61.5 | 61.0 |
| | PA | **25.6** | **33.6** | **47.3** | 43.6 | 47.1 | 46.4 |
| **Hellaswag QA** | | | | | | | |
| Baseline | Acc | 26.5 | 28.8 | 30.6 | 33.3 | 36.1 | 34.6 |
| | PA | 7.5 | 9.4 | 11.8 | 13.9 | 17.3 | 15.0 |
| PRIDE | Acc | 17.8 | 32.7 | 35.6 | 38.6 | 39.6 | 41.4 |
| | PA | 4.9 | 12.9 | 16.0 | 20.5 | 21.8 | 23.2 |
| Attention | Acc | **34.8** | **40.0** | **41.7** | 42.6 | 43.2 | **43.5** |
| score | PA | **18.3** | **22.9** | **24.9** | **27.2** | 26.6 | **26.5** |
| QK-score | Acc | 33.0 | 37.1 | 40.4 | **42.7** | **45.7** | 42.3 |
| | PA | 15.9 | 14.3 | 23.0 | 22.2 | **28.5** | 24.6 |
| **Halu Dialogue** | | | | | | | |
| Baseline | Acc | 21.1 | 30.9 | 34.2 | 36.1 | 34.5 | 35.6 |
| | PA | 5.4 | 10.2 | 14.3 | 18.9 | 16.8 | 20.7 |
| PRIDE | Acc | 3.0 | 32.0 | 35.5 | 36.3 | 36.5 | 36.1 |
| | PA | 0.5 | 12.8 | 18.2 | 17.7 | 18.9 | 20.8 |
| Attention | Acc | 31.4 | **39.9** | 39.3 | 41.1 | 42.1 | 39.9 |
| score | PA | 10.9 | **19.4** | 17.5 | 19.0 | 22.3 | 21.3 |
| QK-score | Acc | **37.1** | 36.6 | **40.6** | **42.3** | **45.3** | **42.8** |
| | PA | **17.7** | 14.6 | **19.6** | **22.0** | **25.9** | **22.2** |

Table 2: Comparison of different methods for LLaMA2-7B (base) on various Q&A datasets. Reported metrics are Accuracy (Acc) and Permutation Accuracy (PA). The best results are highlighted in **bold**.

```python
for index in range(5):
    top_heads[index] = mmlu_top_heads[index] + hellaswag_top_heads[index]

best_heads_across_shots = set(top_heads_for_each_shot[0])

for index in range(1, 5):
    top_heads_for_each_shot &= set(top_heads_for_each_shot[index])
```

Listing 9: Example of calculation of best heads for all 5 shots for two datasets

# E STABILITY OF BEST HEADS

We utilized the minimum accuracy percentiles to determine stable heads that can be seen in Figure 11a. Again, the heads from the 14th layer show the highest accuracy on almost all percentiles. We also listed the top 1% pairs for all setups based on accuracy in Table 9. There is a noticeable

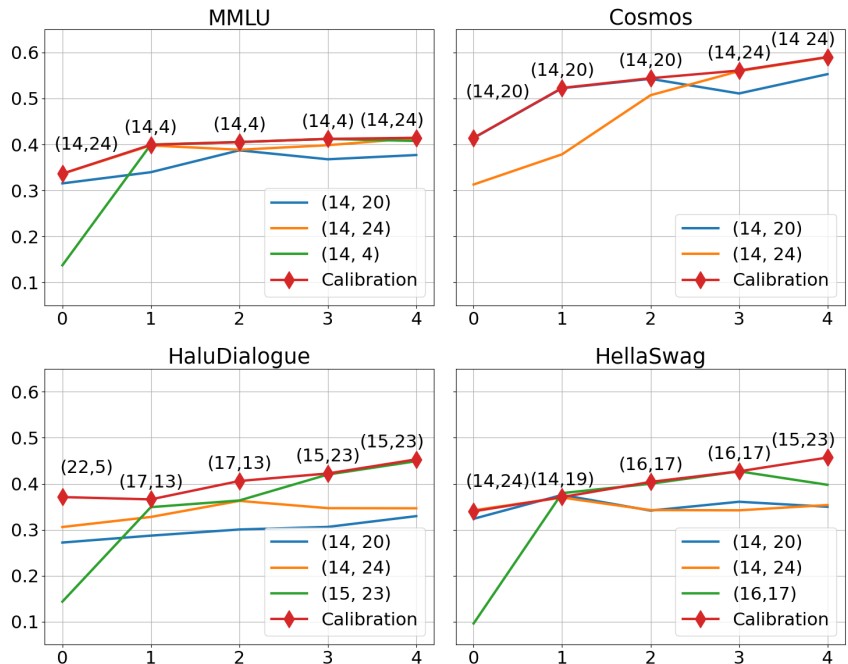

Figure 10: Accuracy of the best performing heads and of the most robust heads — (14, 24), (14, 20)

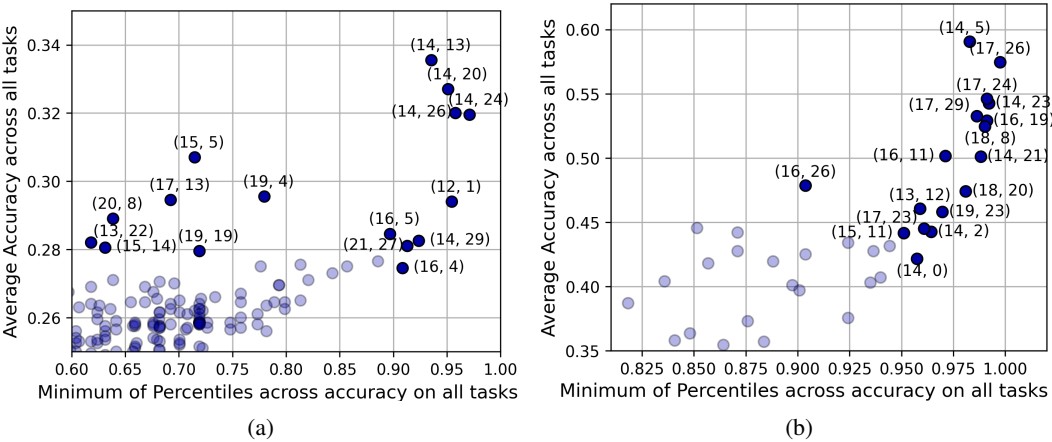

Figure 11: Stable heads for *QK-score* in (a) LLaMA2-7B and (b) LLaMA3-8B for 0-shot setup across all tasks. "$k$-th Minimum of Percentiles" means that the head is better than $k$ share of all heads for all tasks.

overlap between heads for various setups, and, once again, all of them are middle layers of the model.

If we compare the performance of the "stable" heads with results obtained with preceding calibration in Figure 10, (14, 24) and (14, 20) are frequently chosen from the validation set. However, even when they do not, their performance is comparable to that of their validation-chosen counterparts, except for HaluDialogue. Besides, we tested the heads (14, 24), (14, 20), (14, 26), and (14, 13) for stability against increasing the number of options in SSD dataset (see Figure 5b) and against changing the symbols that denote options, following Alzahrani et al. (2024) (see Appendix G). We also added other heads performing well on the SSD dataset to these plots for comparison.

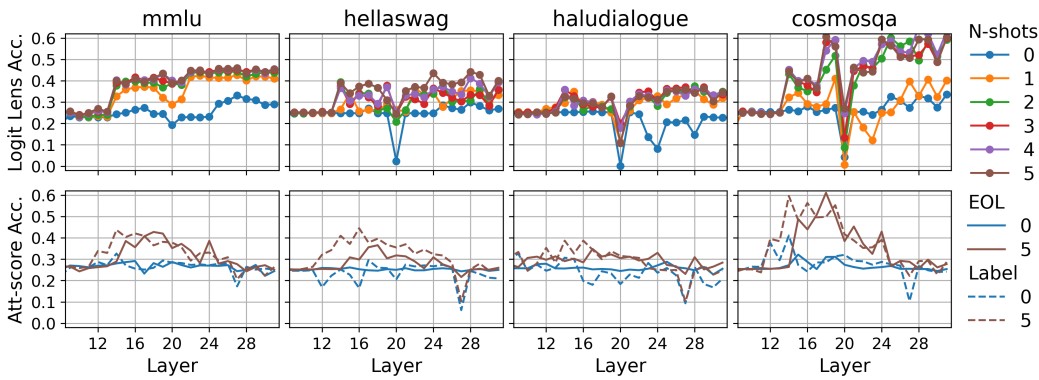

Figure 12: Logit Lens results on LLaMA2-7B base model for 0-shot and few-shot setups (upper) and a comparison to maximal accuracy per layer via *Attention-score* (lower)

## F    LOGIT LENS EXPERIMENT

We follow Halawi et al. (2024) and track the accuracy in the intermediate layers using logit lens (Nostalgebraist, 2020). Denoting $\boldsymbol{h}^{(l)} \in \mathbb{R}^d$ as a hidden state corresponding to last token in layer $l$, we extract intermediate probabilities for options $d_i$ using:

$$P_l(d_i \mid q, d, o) = \text{Logits}^{(l)}_{t_i}, \qquad \text{Logits}^{(l)} = \text{Softmax}(\boldsymbol{W}_U \cdot \text{LayerNorm}(\boldsymbol{h}^{(l)})) \qquad (6)$$

Fig. 12 demonstrates the results for the LLaMA2-7B base model, which shows some interesting patterns. In most cases, we see the improvements after the 12th layer for all setups excluding 0-shot. We see a similar trend as we compare it with the maximal accuracy over *Attention-score* for two different types of option-representative tokens. However, the peak accuracy is seen in the middle layers, after which it degrades. Interestingly, the logit lens performance demonstrates a sudden performance drop of around 20. This indicates that some alternative "thoughts" about the answer emerge at this point, further overlapped by the correct answer.

## G    BEHAVIOUR OF THE BEST HEADS UNDER THE CHANGE OF OPTIONS SYMBOLS AND OPTIONS AMOUNT

Aside from the standard version of the Simple Synthetic Dataset (SSD), which includes four essential options and two additional options, "E" and "F" (described in Section 5.1), we also considered alternative versions of the SSD with varying numbers of possible options. For instance, the version corresponding to the number "10" on the x-axis of Figure 5b contains ten essential options (A, B, C, D, E, F, G, H, I, J) and two special options: "K. I don't know" and "L. None of the above" (see Example 10). In these experiments, we used 200 examples from each version of the dataset to compute the attention scores.

Figure 13 is an extended version of Figure 5b, showing more heads for LLaMA2-7B (left) and a similar experiment for several heads of LLaMA3-8B (right), four of which are taken from the upper right section of Figure 11b, as they are the most stable across real datasets.

```
Which of the following options corresponds to " mediterranean "?
Options:
    A: acceptance
    B: specialties
    C: charitable
    D: typically
    E: access
    F: jose
    G: findlaw
    H: colonial
    I: mediterranean
```

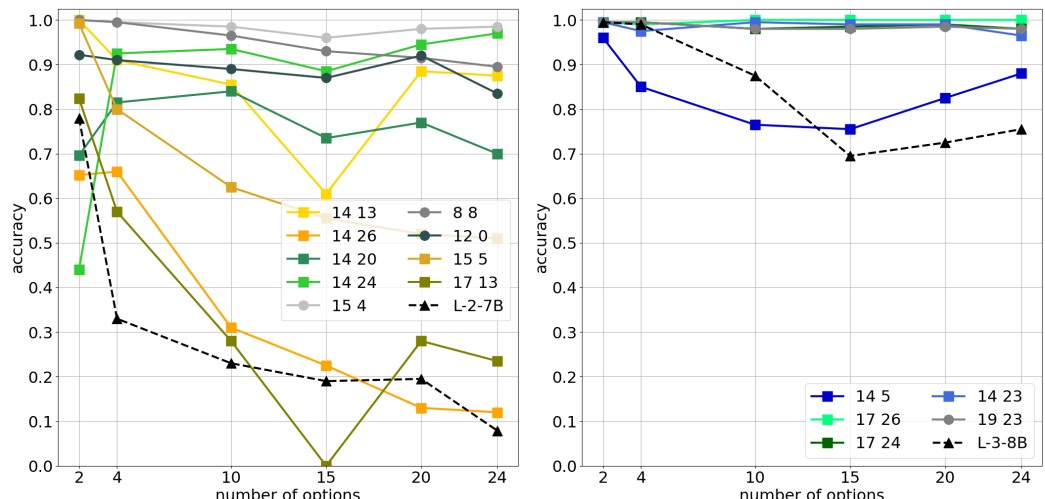

Figure 13: The results for various numbers of options in the Simple Synthetic Dataset (SSD) in a zero-shot setting are shown for LLaMA2-7B (left) and LLaMA3-8B (right). Different line colors represent the QK dot products from different heads. "Square" markers indicate heads that perform well across real datasets, while "round" markers represent heads that perform well on the synthetic dataset.

```
J: data
K: I don't know.
L: None of the above.
```

Listing 10: Modification of SSD with ten options - example

In Figure 14, we return to the standard 4-option SSD dataset but use different symbols for the option labels. The upper plot includes the renamed special options "E" and "F", while the lower plot omits them for the LLaMA2-7B model. Similarly, Figure 15 shows the same setup for the LLaMA3-8B model.

# H COMPREHENSIVE RESULTS FOR EXPERIMENTS ON LARGER MODELS FROM LLAMA FAMILY

Here, we provide complete results of our experiments with QK-scores on four primary datasets (MMLU, CosmosQA, HellaSwag, and Halu Dialogue) for larger models. As before, the reported metrics are Accuracy and Permutation Accuracy.

- Figure 17 contains results for LLaMA2-13B, and Figure 23 for its chat-tuned version
- Figure 18 contains results for LLaMA2-70B, and Figure 24 for its chat-tuned version
- Figure 19 contains results for LLaMA3-8B, and Figure 25 for its instruct-tuned version
- Figure 20 contains results for LLaMA3-70B, and Figure 26 for its instruct-tuned version
- Figure 21 contains results for LLaMA-30B
- Figure 22 contains results for LLaMA-65B.

Our experiments' accuracy scores for these baseline models are somewhat lower than those in the original technical reports (et. al., 2023; Dubey et al., 2024). The main reason for this is that we added additional "E" and "F" options not used in those reports; some differences in prompts and particular examples for few-shot learning could also play a role. Also note that in many experiments we focus on zero-shot scenario without chain-of-thoughts prompting, which received less attention in the original technical reports.

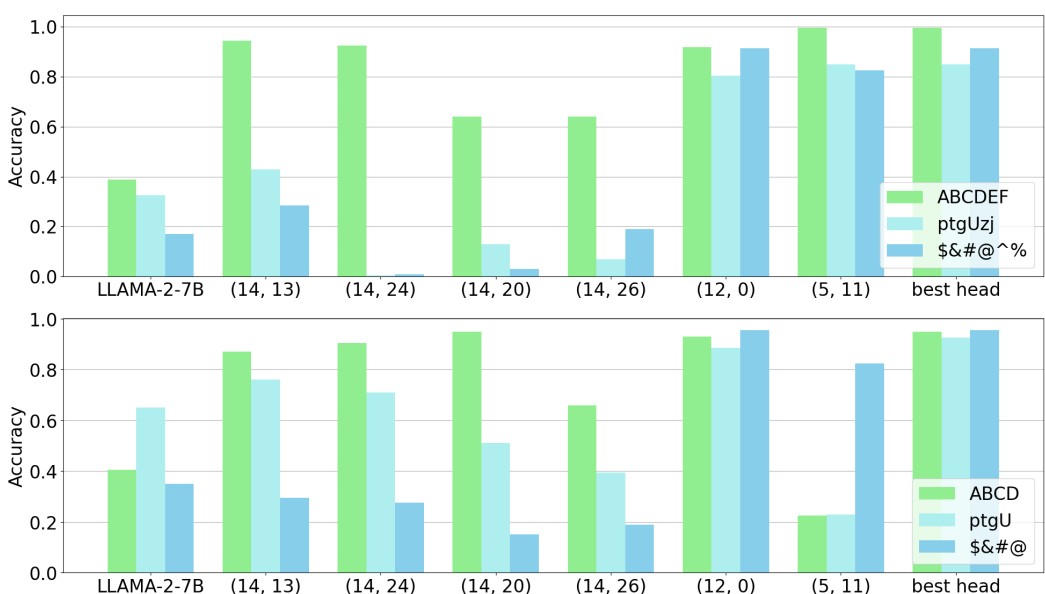

Figure 14: Performance of the QK-score from the best heads of the LLaMA2-7B model for different option symbols, with "uncertainty" options (i.e. "I don't know" and "None of the above") presented (upper figure) and not presented (lower figure). The accuracy of the best four heads from Figure 11a declines in these new setups, but the head (12, 0) remains stable across all setups. Another interesting head is (5, 11): its accuracy is high for all setups with "uncertainty" and for "$&#" setup, but drops abruptly for "ABCD" and "ptgU". Studying such "anomalies" is a subject for future research.

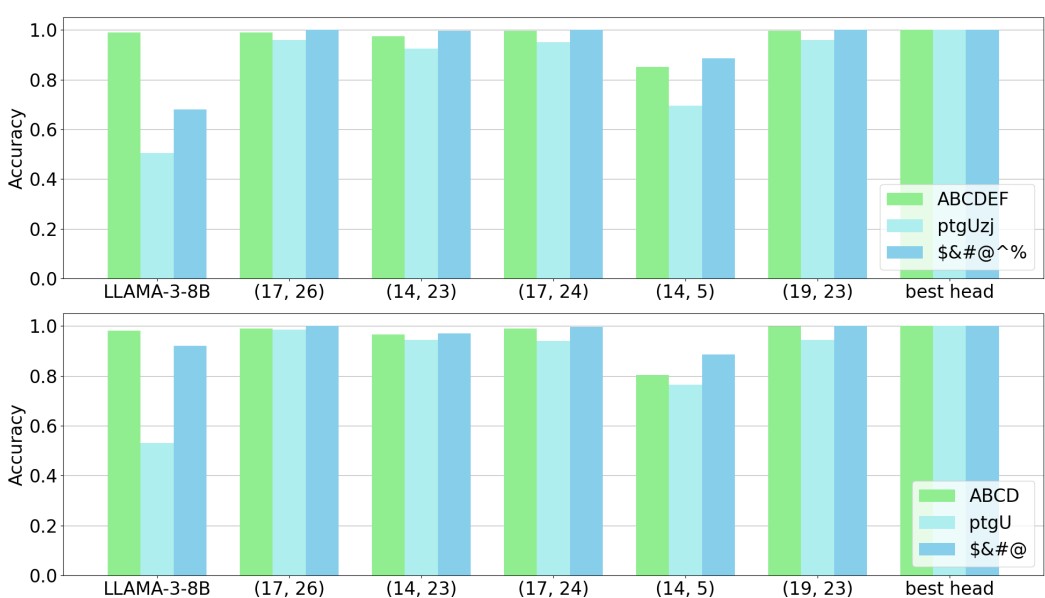

Figure 15: Performance of the QK-score from the best heads of the LLaMA3-8B model for different option symbols, with and without "uncertainty" options. Interestingly, the best heads of the LLaMA3-8B model (see Figure 11b) are significantly more stable across the considered setups.

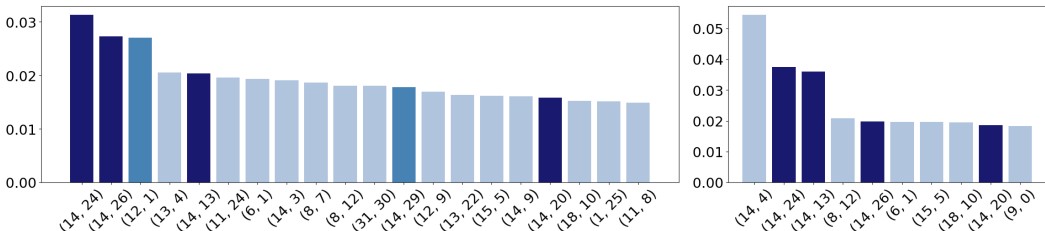

Figure 16: Left: average top heads scores across real datasets (first twenty). Dark blue marks four heads from the top-right corner of Figure 11a. Medium blue marks other heads with accuracy percentiles above 0.9 across all tasks. As shown, the first two heads with the best scores across real datasets belong to the group in the top-right corner of Figure 11a. Right: Top head scores on the Simple Synthetic Dataset (first ten). The top-scored head (14, 4) does not appear in the top-right corner of Figure 11a, but it is listed in Figure 10 as one of the best heads for the MMLU dataset. Note that for calculating this score, we did not use the dataset labels.

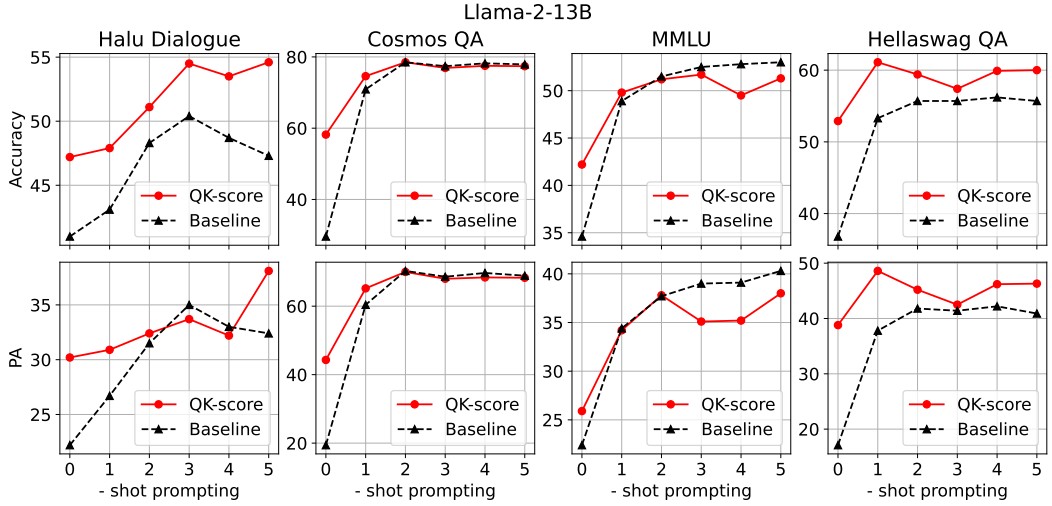

Figure 17: Comparison of different methods for LLaMA2-13B (base) on various Q&A datasets.

## I  HEAD SCORING WITHOUT VALIDATION SET

Let $\hat{\mathcal{D}}$ be some unlabelled MCQA dataset. Then, for each head we may calculate a score

$$HeadScore = \left( \frac{1}{|\hat{\mathcal{D}}|} \sum_{\hat{\mathcal{D}}} \sum_{i=1}^{n} a_{Nt_i} \right) \left( \frac{1}{|\hat{\mathcal{D}}|} \mathbb{I}\{\arg \max_i (a_{Nt_i}) \neq \hat{i}\} \right),$$

where $\hat{i}$ denotes the most frequent option for the given head; head indices $(l, h)$ are omitted. The left component represents the average amount of attention concentrated on the option-representative tokens $t_i, i = 1, \ldots, n$. The right component reflects the frequency of the situation, when the largest attention among the options falls on the option other than $\hat{i}$, i.e., any option other than the most frequent one.

The results of ranking heads according to these scores are presented in Figure 16.

## J  SELECTION BIAS

We investigate our methods towards the tendency to choose a specific option instead of a correct answer. Fig. 40 presents a selection bias for baseline and 3 heads for *QK-score* in 0-shot and 5-shot regimes.

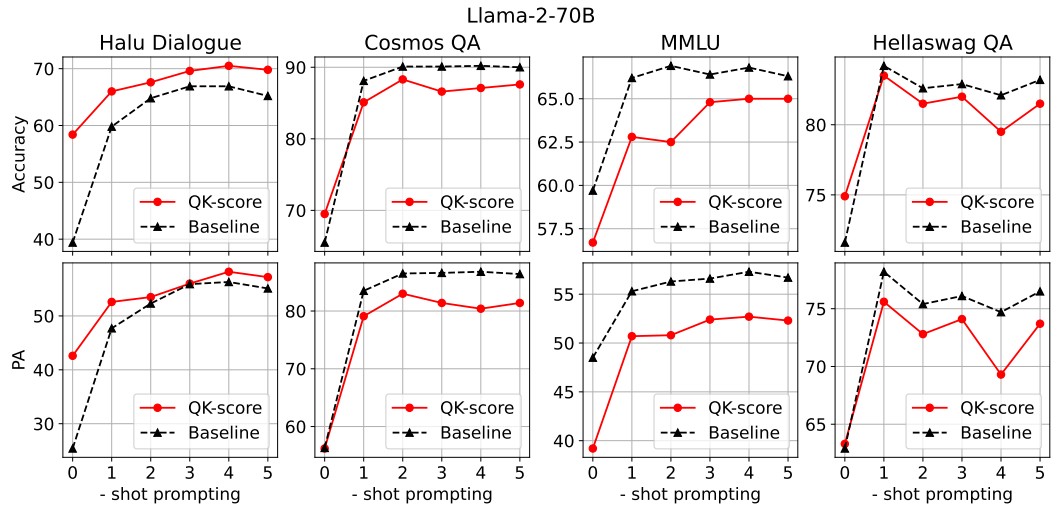

Figure 18: Comparison of different methods for LLaMA2-70B (base) on various Q&A datasets.

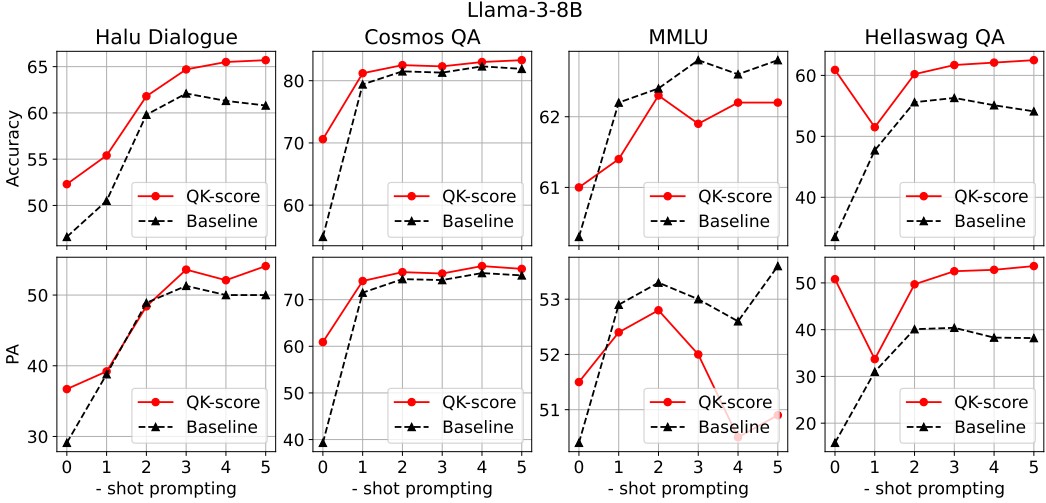

Figure 19: Comparison of different methods for LLaMA3-8B (base) on various Q&A datasets.

Table 3 shows the selection bias regarding recall. We can see that most methods (especially in a 0-shot setup) concentrate on single or several options.

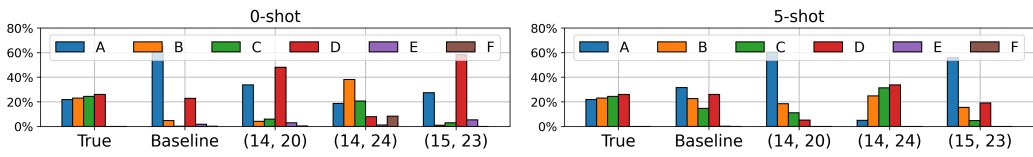

Figure 40: Distribution of predictions across options for different methods on MMLU 0-shot (upper) and 5-shot (lower) setup. $(l, h)$ depicts the distribution for $S_{QK}^{(l,h)}$

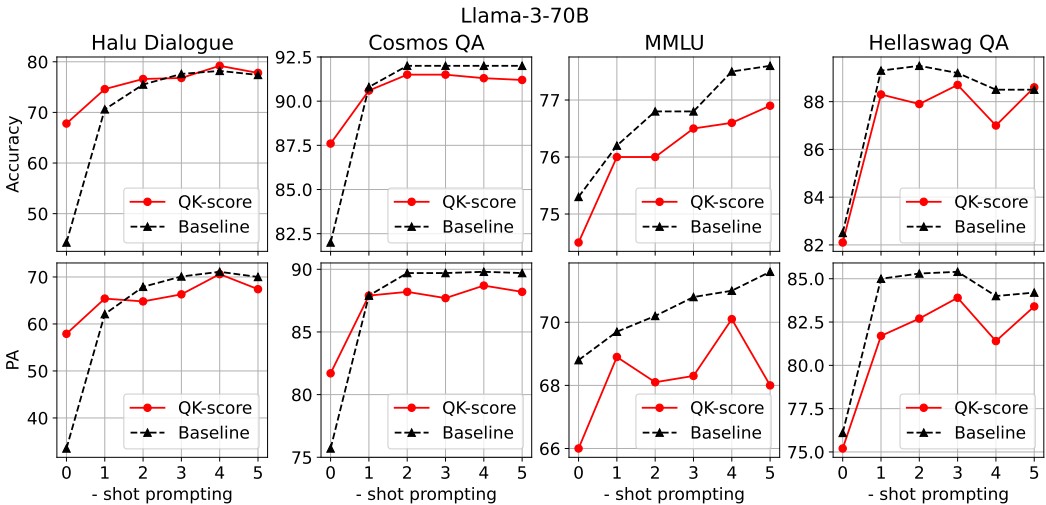

Figure 20: Comparison of different methods for LLaMA3-70B (base) on various Q&A datasets.

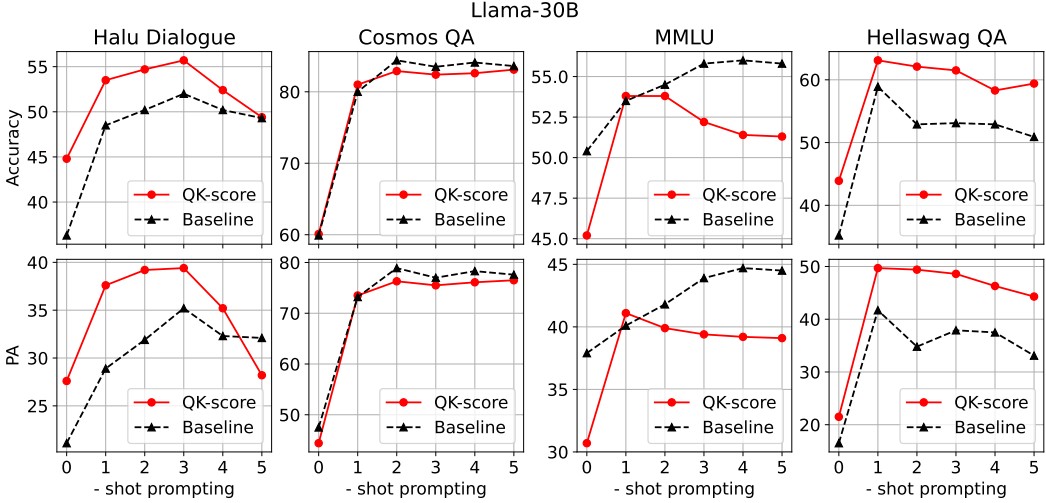

Figure 21: Comparison of different methods for LLaMA-30B (base) on various Q&A datasets.

| Method | 0-shot | | | | | 5-shot | | | | |
|---|---|---|---|---|---|---|---|---|---|---|
| | Orig. | A | B | C | D | Orig. | A | B | C | D |
| Baseline | 26.6 | 73.4 | 7.9 | 0.6 | 28.3 | 43.9 | 41.7 | 53.4 | 38.2 | 42.5 |
| | | (+46.8) | (-18.7) | (-26.0) | (+1.7) | | (-2.2) | (+9.5) | (-5.7) | (-1.4) |
| $S_{QK}^{(14,20)}$ | 31.4 | 43.9 | 9.0 | 10.8 | 60.2 | 37.6 | 78.2 | 40.5 | 25.2 | 12.4 |
| | | (+12.5) | (-22.4) | (-20.6) | (+28.8) | | (+40.6) | (+2.9) | (-12.4) | (-25.2) |
| $S_{QK}^{(14,24)}$ | 33.6 | 30.7 | 58.5 | 33.2 | 14.4 | 43.0 | 14.3 | 49.7 | 53.0 | 51.9 |
| | | (-2.9) | (+24.9) | (-0.4) | (-19.2) | | (-28.7) | (+6.7) | (+10.0) | (+8.9) |
| $S_{QK}^{(15,23)}$ | 26.2 | 30.9 | 2.1 | 5.0 | 63.6 | 36.3 | 71.2 | 36.3 | 14.7 | 27.0 |
| | | (+4.7) | (-24.1) | (-21.2) | (+37.4) | | (+34.9) | (+0.0) | (-21.6) | (-9.3) |

Table 3: Selection bias for different methods on MMLU 0-shot and 5-shot using LLaMA2-7B. The table compares original accuracy (for the task to predict A/B/C/D/E/F) and recall only on the subset with single ground truth option (i.e., only questions with answer A).

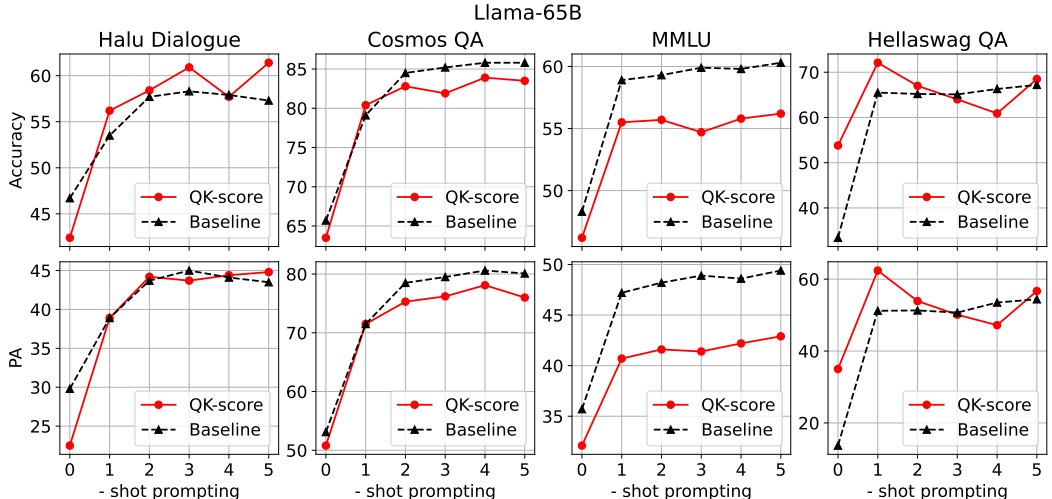

Figure 22: Comparison of different methods for LLaMA-65B (base) on various Q&A datasets.

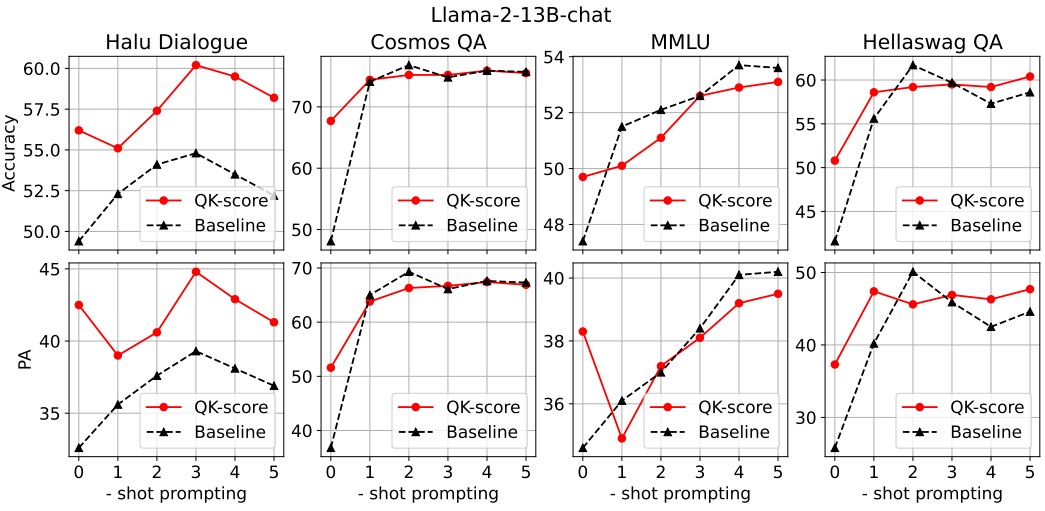

Figure 23: Comparison of different methods for LLaMA2-13B-chat on various Q&A datasets.

## K   SYNTHETIC DATASET IN DIFFERENT LANGUAGES

We regenerated our synthetic dataset using three languages in addition to English. Figure 41 shows that the general distribution of QK-scores across heads of the LLaMA2-7B model on these multilingual datasets remains largely unchanged; for example, layers 8–15 still contain the most performant heads. However, differences in the performance of individual heads are also observed.

Below are the top-10 best-performing heads for each language, sorted by decreasing accuracy:

- EN: **(8, 8)**, **(15, 4)**, (12, 15), **(14, 24)**, (12, 10), (14, 13), **(14, 27)**, **(12, 25)**, **(12, 21)**, **(14, 20)**, accuracy decreasing from 0.995 to 0.815;

- IT: **(12, 21)**, **(15, 4)**, **(8, 8)**, **(14, 20)**, (12, 13), **(14, 24)**, **(14, 27)**, (12, 0), **(12, 25)**, (12, 10), accuracy decreasing from 1.0 to 0.9;

- FR: **(12, 21)**, (12, 13), **(8, 8)**, **(14, 20)**, **(15, 4)**, **(14, 27)**, **(14, 24)**, (8, 21), **(12, 25)**, (12, 0), accuracy decreasing from 1.0 to 0.905;

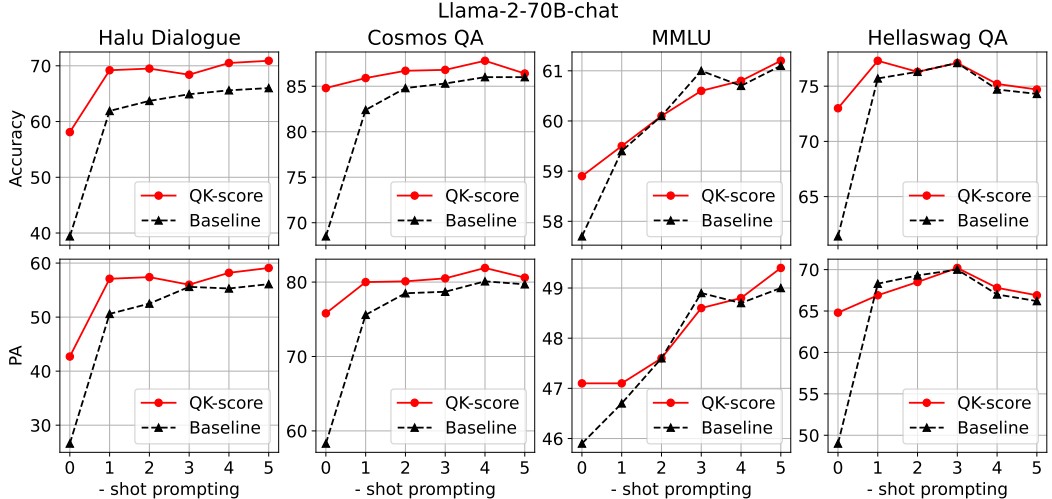

Figure 24: Comparison of different methods for LLaMA2-70B-chat on various Q&A datasets.

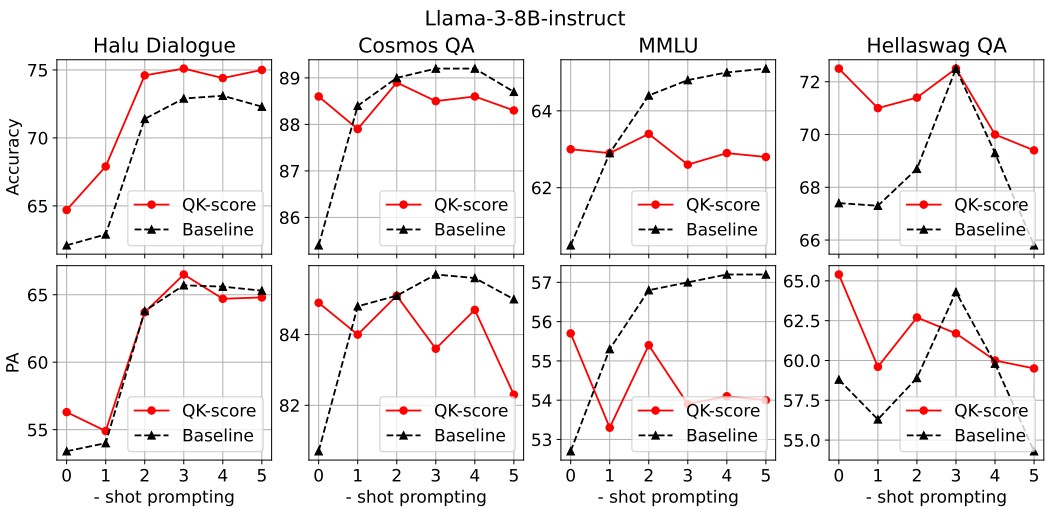

Figure 25: Comparison of different methods for LLaMA3-8B-instruct on various Q&A datasets.

- RU: **(12, 25)**, (14, 20), **(8, 8)**, (12, 13), (12, 15), **(12, 21)**, **(15, 4)**, (14, 24), **(14, 27)**, (12, 6), accuracy decreasing from 1.0 to 0.91.

We colored green the heads that perform best across four real datasets (see Figure 5a). Additionally, we highlighted in bold the heads that appear in the top-10 for all four languages.

As shown, 7 out of the top-10 best heads are shared across synthetic datasets in different languages, including two "green" heads that are also the best across our real datasets. This significant overlap suggests a substantial degree of universality among the identified heads. Interestingly, the QK-scores for the best heads are somewhat lower for English compared to the other languages we analyzed. However, we cannot draw definitive conclusions from this observation without further investigation. A more thorough study of how QK-scores and the best-performing heads vary with the dataset's language remains a topic for future research and is beyond the scope of this paper.

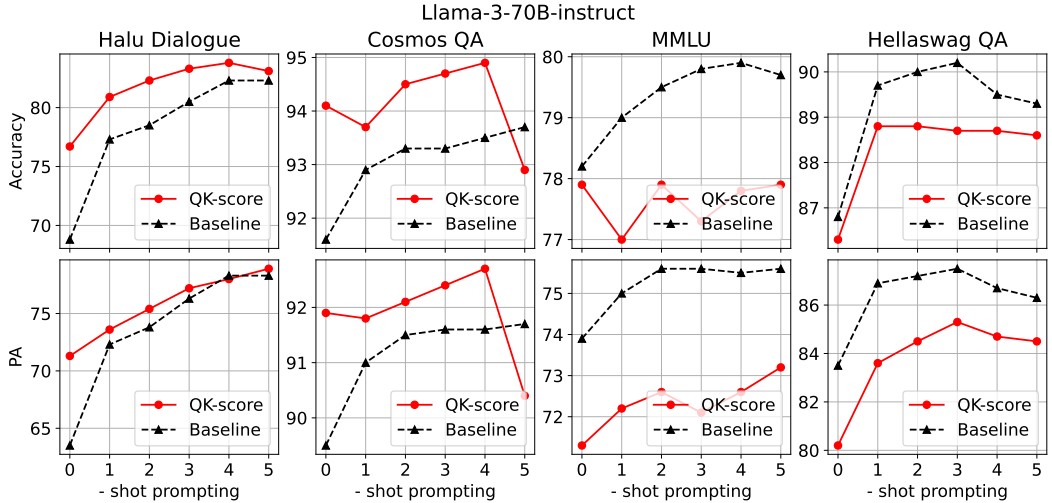

Figure 26: Comparison of different methods for LLaMA3-70B-instruct on various Q&A datasets.

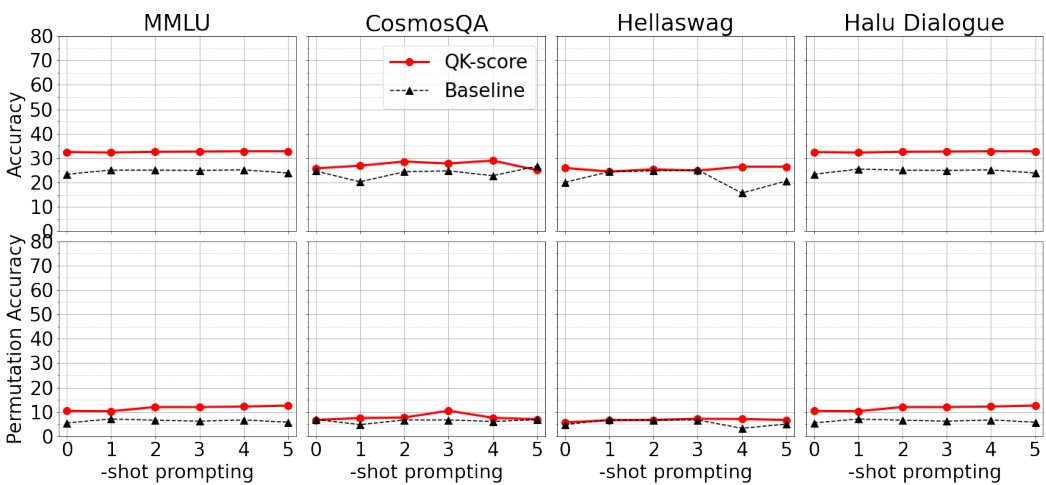

Figure 27: Comparison of different methods for Dolly V2-3B on various Q&A datasets.

## L  COMPREHENSIVE RESULTS FOR EXPERIMENTS ON QWEN-2.5 AND OTHER MODEL FAMILIES

Here, we present the results of our experiments with QK-scores on four main datasets (MMLU, CosmosQA, HellaSwag, and Halu Dialogue) for models from other families. As in previous experiments, the reported metrics are Accuracy and Permutation Accuracy.

- Figure 27 contains results for Dolly V2-3B
- Figure 28 contains results for Gemma-2B
- Figure 29 contains results for Phi-3.5-Instruct
- Figure 30 contains results for Qwen-2.5-1.5B, and Figure 31 for its instruct-tuned version
- Figure 32 contains results for Qwen-2.5-7B, and Figure 33 for its instruct-tuned version
- Figure 34 contains results for Qwen-2.5-14B, and Figure 35 for its instruct-tuned version.
- Figure 36 contains results for Qwen-2.5-32B, and Figure 37 for its instruct-tuned version.
- Figure 38 contains results for Qwen-2.5-72B, and Figure 39 for its instruct-tuned version.

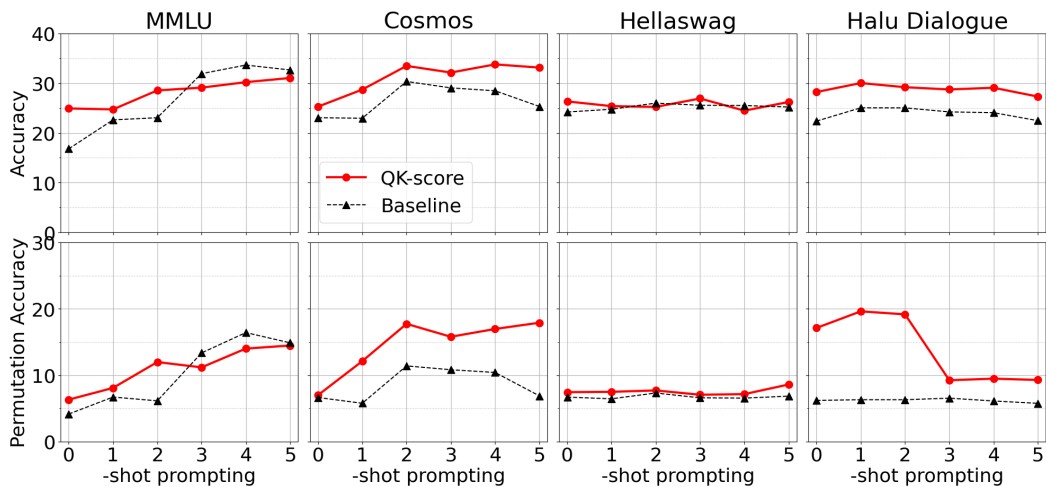

Figure 28: Comparison of different methods for Gemma-2B on various Q&A datasets.

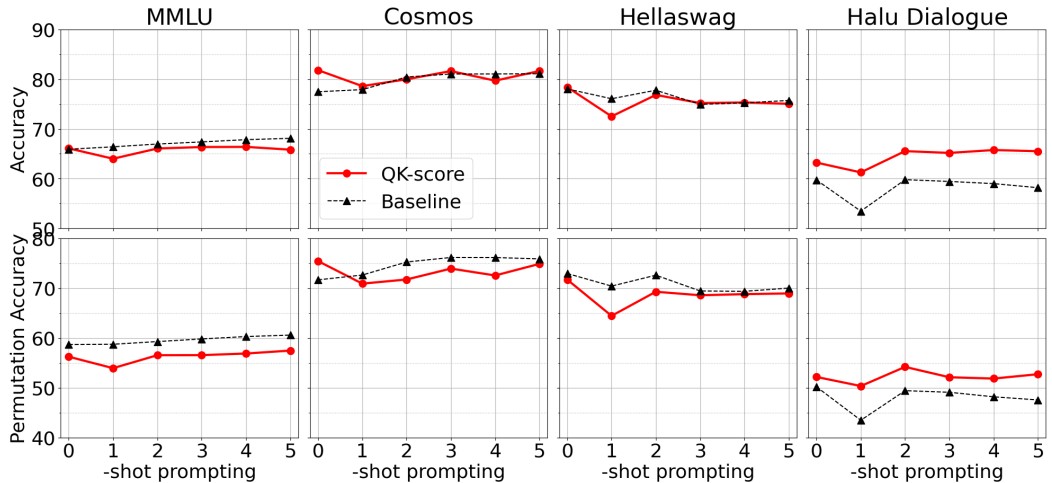

Figure 29: Comparison of different methods for Phi-3.5-mini (instruct tuned) on various Q&A datasets.

We observe that the accuracy and permutation accuracy plots for the baseline and QK-scores of models around 3B in size — specifically, Dolly-v2-3B (Figure 27) and Phi-3.5-mini (Figure 29) - mostly show behavior similar to LLaMA-2 models ranging from 7B (Figure 3) to 13B (Figure 17). For these models, the QK-score is usually higher than the baseline score in zero-shot setups, and the QK-score and baseline often show convergence in few-shot setups, though at times they remain at a similar distance from each other. A similar trend is observed for the Qwen 2.5 models, with sizes of 7B and 14B, especially for instruct versions. The QK-score is also typically better than the baseline for Gemma-2B (Figure 28), although the plots for this model exhibit some unusual patterns in certain cases.

However, we observe a deviation from the general trend for both the base and instruct versions of the smallest model, Qwen 2.5-1.5B. Specifically, the QK-score and baseline scores are unusually close to each other in few-shot setups, and in several cases, the QK-score performs worse than the baseline in zero-shot setups. We hypothesize that this may be due to these models being overly fine-tuned for multiple-choice question answering (MCQA), which alters the baseline behavior in zero-shot setups. This hypothesis is supported by the fact that these models outperform LLaMA2-7B and other larger models in baseline setups.

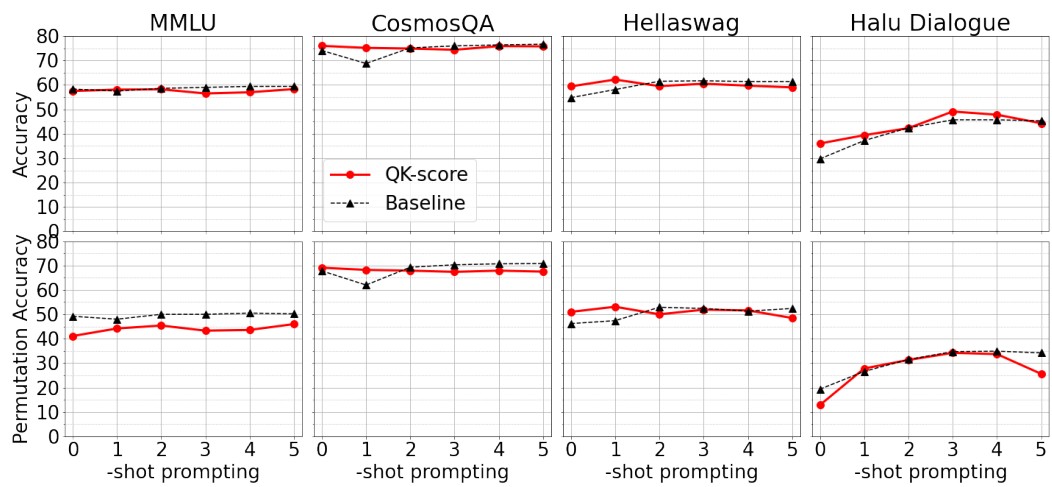

Figure 30: Comparison of different methods for Qwen-1.5B on various Q&A datasets.

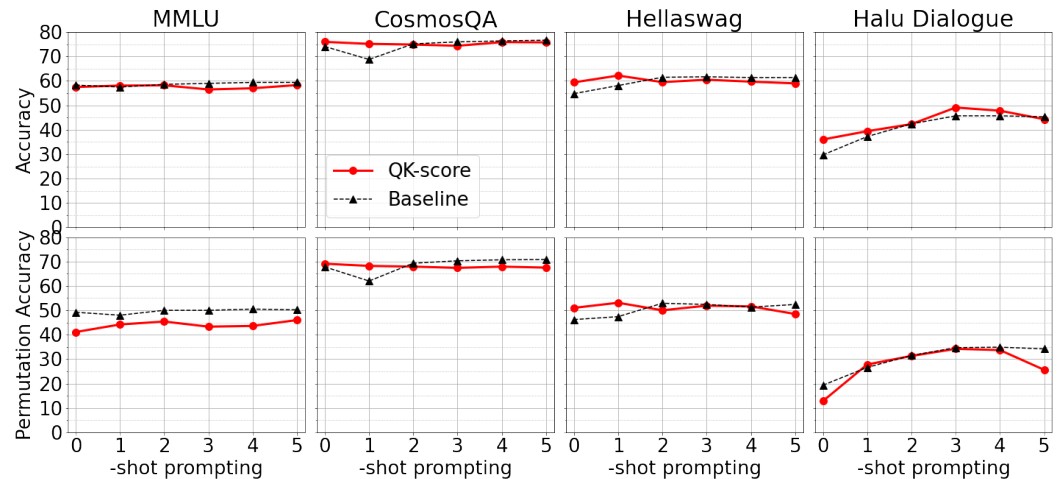

Figure 31: Comparison of different methods for Qwen-1.5B-Instruct on various Q&A datasets.

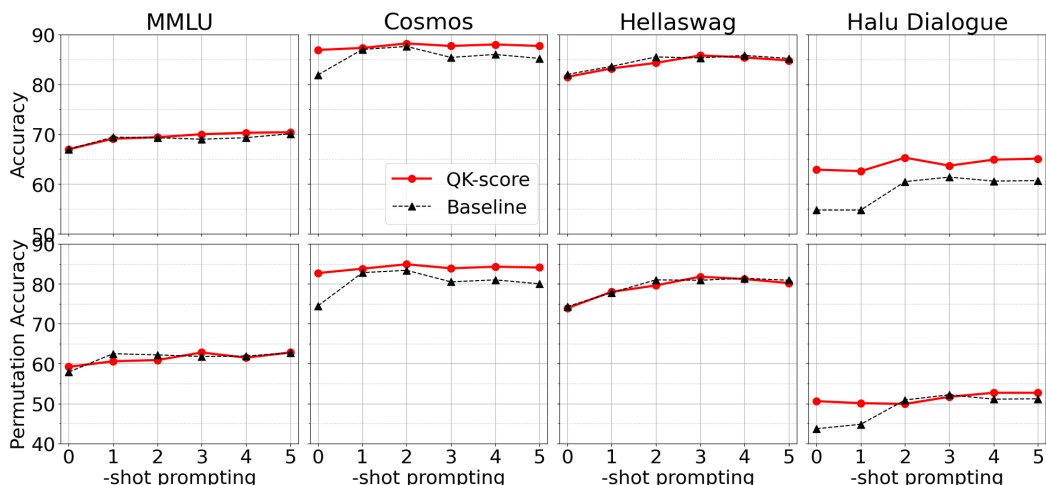

Figure 32: Comparison of different methods for Qwen-7B on various Q&A datasets.

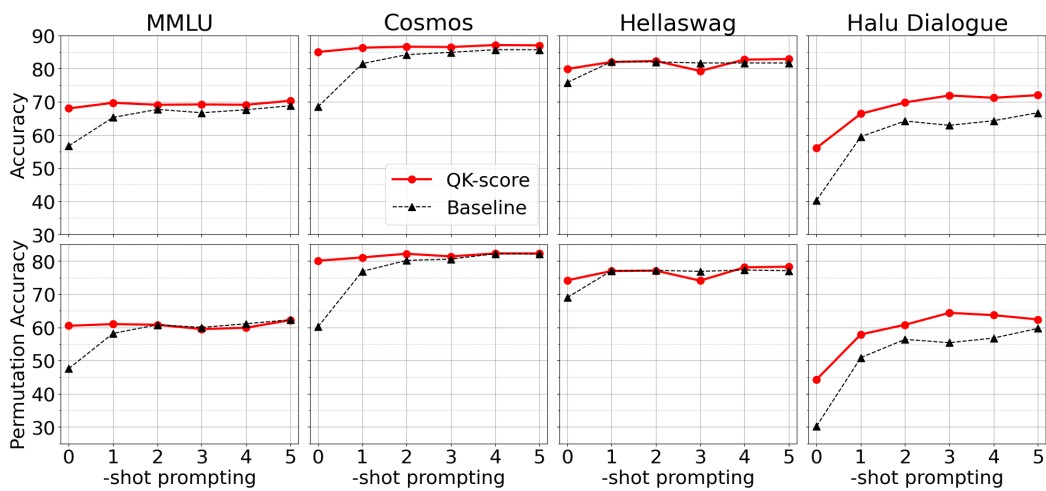

Figure 33: Comparison of different methods for Qwen-7B-Instruct on various Q&A datasets.

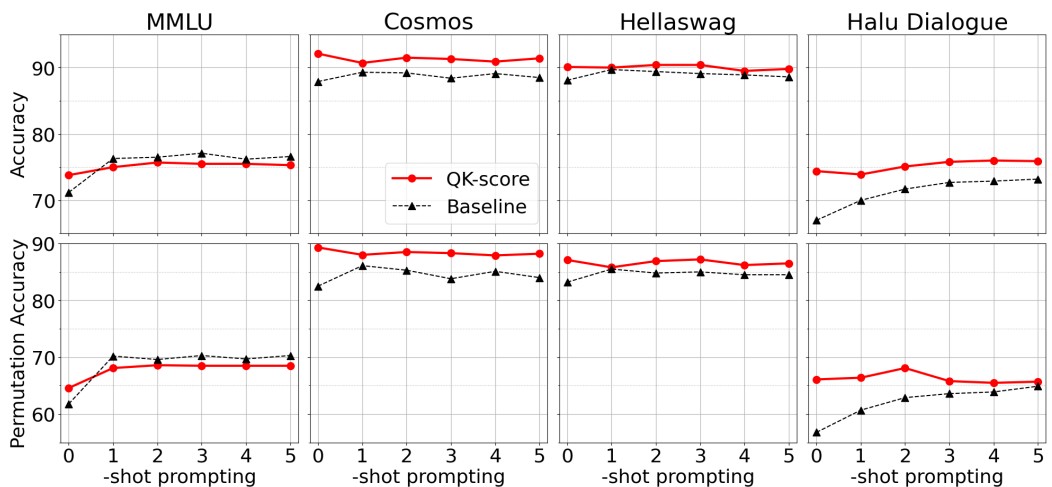

Figure 34: Comparison of different methods for Qwen-14B on various Q&A datasets.

Intrigued by these differences, we conducted an additional analysis of the behavior of individual heads in Qwen 2.5-1.5B. We confirmed that Qwen 2.5-1.5B-base contains several heads that consistently perform well across both real and synthetic datasets, such as heads **(20, 4)** and **(21, 11)**. In this regard, it remains similar to the LLaMA models. For a detailed discussion of individual head performance on the SSD dataset, see Appendix M.

## M  BEST HEADS ON SYNTHETIC DATASET FOR QWEN 2.5-1.5B

In Figures 42a and 42b, we present the accuracy of the QK-score for each head of Qwen 2.5-1.5B-base and -instruct on our synthetic dataset. These diagrams confirm that the layers with the best heads in Qwen 2.5 are closer to the final layer than in LLaMA-family models. Specifically, the best-performing heads in Qwen 2.5 are concentrated in layers 16–22, out of a total of 28 layers. Besides, earlier layers contain many heads with performance close to zero. Despite these differences, the overall pattern resembles the corresponding heatmap for LLaMA-2-7B shown in Figure 41a, particularly in that very early and very late layers do not contain strongly pronounced select-and-copy heads. We also observe that the heads **(20, 4)** and **(21, 11)**, which perform well across real datasets in Qwen 2.5-1.5B-base, remain among the best heads for the synthetic dataset. This indicates that some heads are persistent and perform well across multiple datasets, as discussed in our paper.

# N    RESULTS FOR QK-SCORE ON FINE-TUNED QA MODELS

We have finetuned the LLaMA-2-7B on each dataset and tested how our method performs after fine-tuning. We trained LoRA adapters and merged them with the model. The results are in the Table 4. To test the models, we used the subset of data the model did not seen during the train.

| | | MMLU | | CosmosQA | | HaluDialogue | | HellaSwag | |
|---|---|---|---|---|---|---|---|---|---|
| | | SFT | LLaMA | SFT | LLaMA | SFT | LLaMA | SFT | LLaMA |
| 0-shot | Baseline | 0.493 | 0.267 | 0.863 | 0.311 | 0.923 | 0.211 | 0.352 | 0.265 |
| | QK | 0.488 | 0.336 | 0.840 | 0.414 | 0.905 | 0.371 | 0.393 | 0.330 |
| 1-shot | Baseline | 0.476 | 0.391 | 0.836 | 0.393 | 0.474 | 0.309 | 0.716 | 0.288 |
| | QK | 0.472 | 0.407 | 0.810 | 0.5 | 0.858 | 0.366 | 0.800 | 0.371 |
| 2-shot | Baseline | 0.477 | 0.431 | 0.643 | 0.591 | 0.823 | 0.342 | 0.795 | 0.306 |
| | QK | 0.477 | 0.421 | 0.685 | 0.615 | 0.872 | 0.406 | 0.803 | 0.404 |
| 3-shot | Baseline | 0.483 | 0.437 | 0.670 | 0.569 | 0.621 | 0.361 | 0.681 | 0.33 |
| | QK | 0.470 | 0.405 | 0.692 | 0.593 | 0.867 | 0.423 | 0.786 | 0.427 |
| 4-shot | Baseline | 0.478 | 0.441 | 0.820 | 0.579 | 0.730 | 0.345 | 0.726 | 0.361 |
| | QK | 0.470 | 0.420 | 0.802 | 0.615 | 0.877 | 0.453 | 0.789 | 0.457 |
| 5-shot | Baseline | 0.487 | 0.438 | 0.827 | 0.547 | 0.746 | 0.356 | 0.678 | 0.346 |
| | QK | 0.482 | 0.427 | 0.810 | 0.61 | 0.880 | 0.428 | 0.786 | 0.423 |

Table 4: Accuracy on supervised fine-tuned LLaMA2-7B on the same dataset the model was fine-tuned

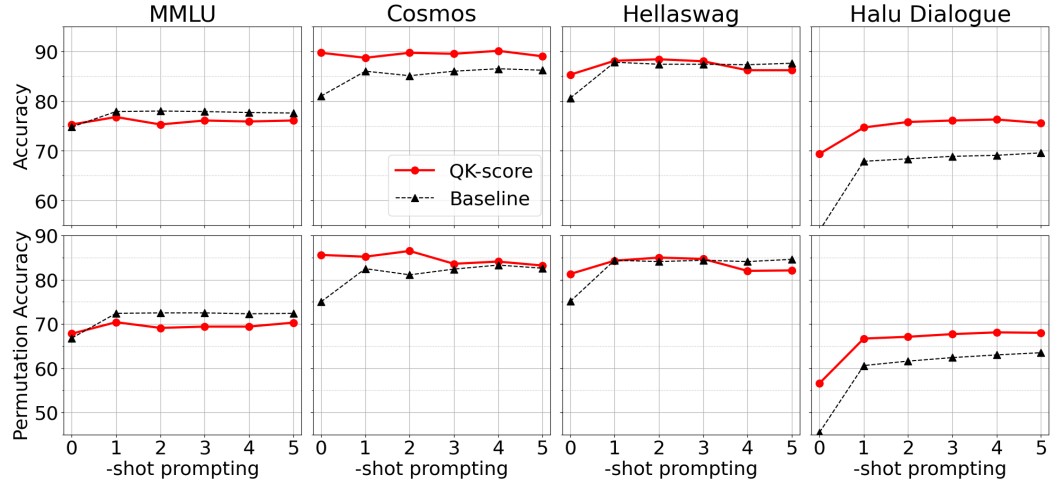

Figure 35: Comparison of different methods for Qwen-14B-Instruct on various Q&A datasets.

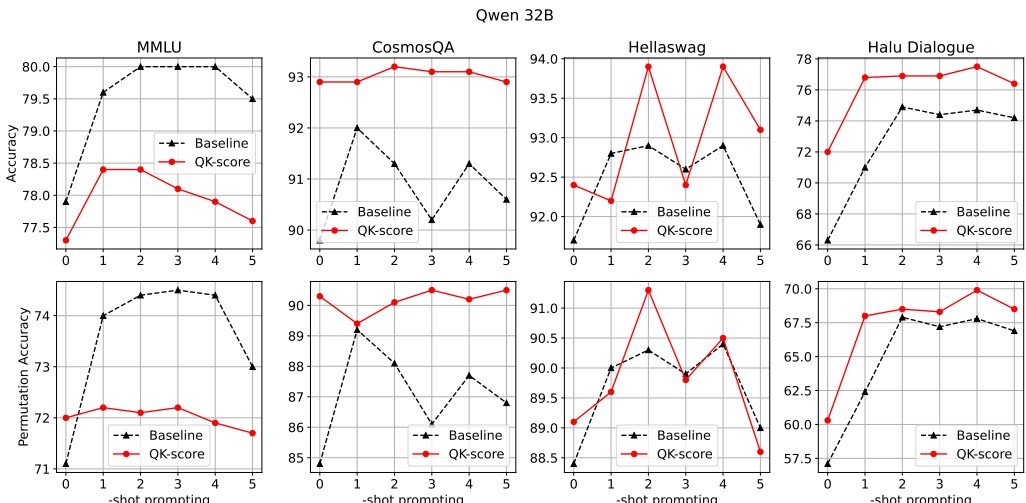

Figure 36: Comparison of different methods for Qwen-32B on various Q&A datasets.

## O RESULTS FOR CLOZE PROMPTING

Cloze-style evaluation (or cloze prompting)(Robinson & Wingate, 2023) has been widely used for evaluating language models, but it has certain drawbacks. These include the "probability stealing" effect, where the correct answer's probability is spread across different surface forms(Wiegreffe et al., 2023),(Alzahrani et al., 2024). Cloze prompting is also sensitive to prompt phrasing and may lead to overfitting to training patterns. Although multi-choice prompting (MCP) addresses some of these issues, it introduces its own biases, such as position and label biases, and is sensitive to sample order in few-shot settings. Additionally, smaller models often struggle with the required output format(Alzahrani et al., 2024), (Khatun & Brown, 2024).

As large language models (LLMs) have advanced, the format of Question Answering tasks has shifted from cloze prompting to multiple-choice formulations (Gu et al., 2024), (OpenAI, 2024), which aligns with the focus of this study. By addressing the limitations of both cloze and MCQA prompting, our method aims to provide a more reliable and insightful model evaluation.

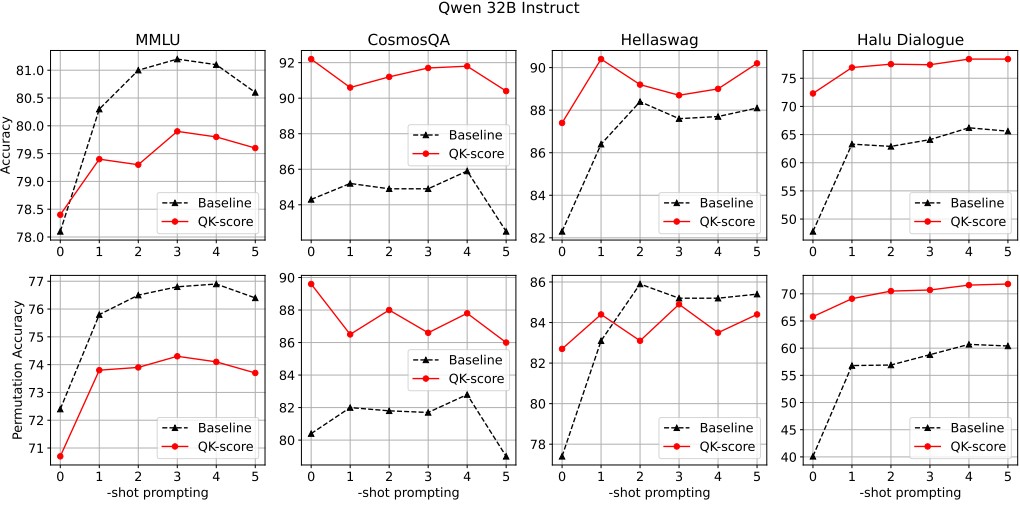

Figure 37: Comparison of different methods for Qwen-32B-Instruct on various Q&A datasets.

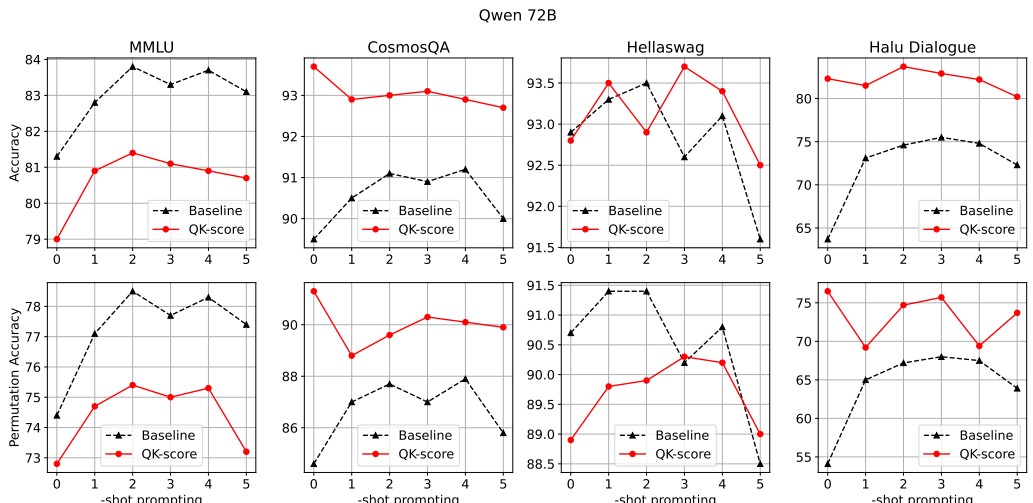

Figure 38: Comparison of different methods for Qwen-72B on various Q&A datasets.

Our method addresses several of the issues mentioned above by separating option selection from text generation within the language model. Compared to cloze and multi-choice prompting, our approach is less sensitive to answer format and wording. It also reduces common biases in MCP, such as those related to option position or the label, by disregarding the most biased attention heads. Due to the differing nature of biases, our method and cloze prompting offer complementary insights. We plan to explore how these two methods can be combined in future work.

A comparison of cloze prompting and our method is presented in Table 5. We observe that QK performs similarly to cloze prompting on the MMLU dataset, which requires short, knowledge-based answers, with only slight degradation in the zero-shot setting. The same holds for CosmosQA, where the model is expected to answer questions based on context. However, for datasets that assess the model's ability to continue given text snippets, QK significantly underperforms. This finding aligns with our understanding of QK as a method that separates semantic decision-making from text generation. On datasets like HellaSwag and HaluDialogue, the correct answer is often determined by the consistency of the text snippet rather than by factual accuracy or commonsense reasoning. We believe that the QK-score is more heavily influenced by the latter.

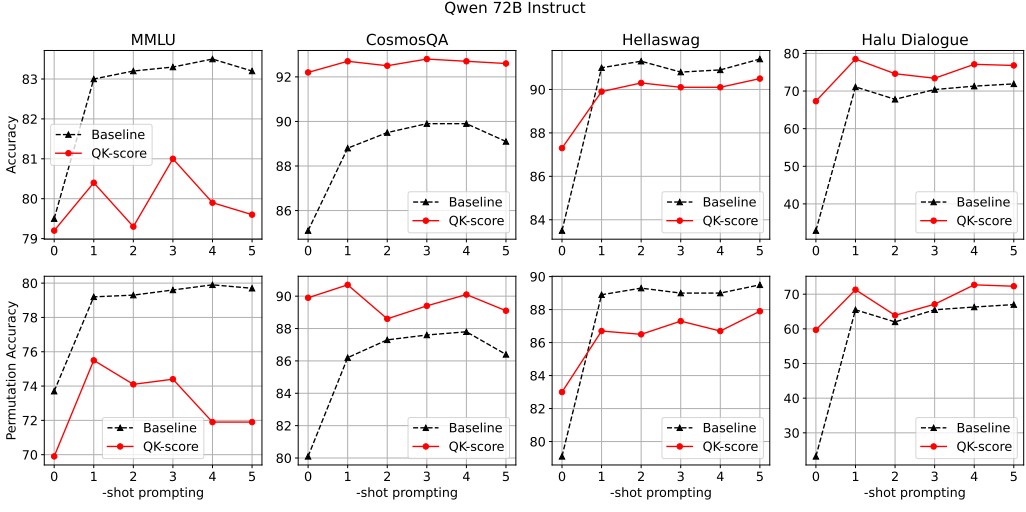

Figure 39: Comparison of different methods for Qwen-72B-Instruct on various Q&A datasets.

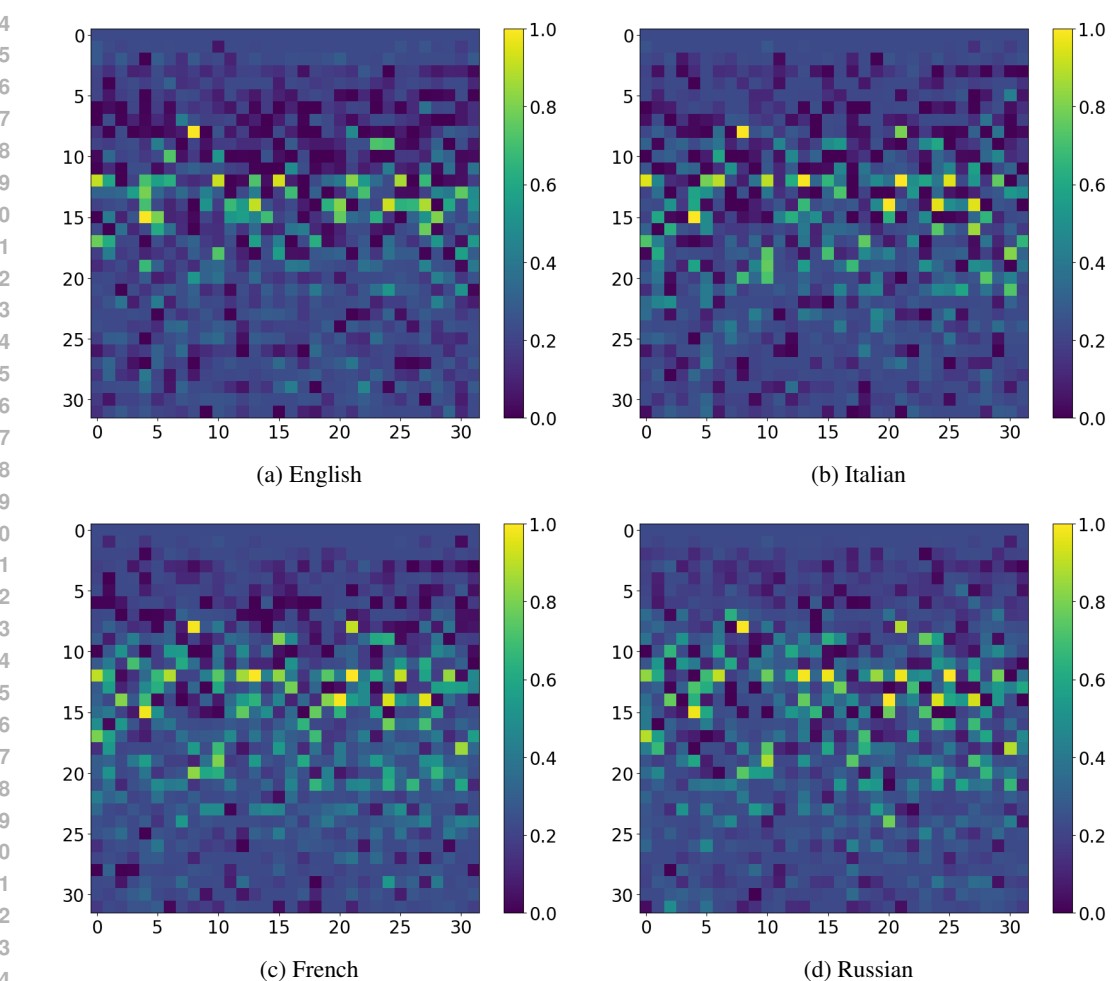

Figure 41: Performance of QK-score across different heads of LLAMA-2-7B on a synthetic dataset generated in multiple languages

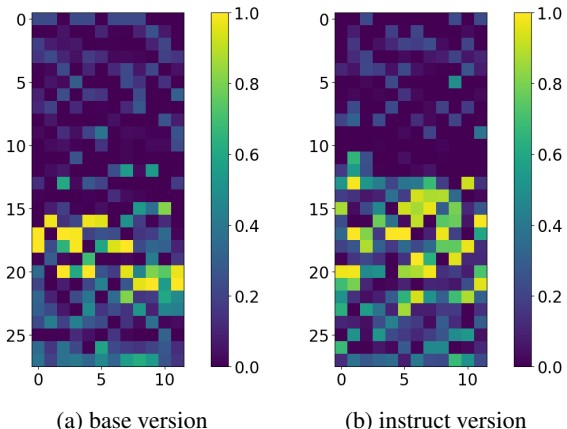

Figure 42: Performance of QK-score across different heads of Qwen 2.5-1.5B on a synthetic dataset

In summary, our approach achieves comparable performance while offering complementary insights into model behavior, making it a valuable alternative to traditional cloze prompting.

| | MMLU | | CosmosQA | | HaluDialogue | | HellaSwag | |
|---|---|---|---|---|---|---|---|---|
| | Cloze | QK | Cloze | QK | Cloze | QK | Cloze | QK |
| 0-shot | 0.38 | 0.35 | 0.49 | 0.46 | 0.42 | 0.40 | 0.52 | 0.38 |
| 1-shot | 0.40 | 0.39 | 0.51 | 0.50 | 0.45 | 0.42 | 0.52 | 0.35 |
| 2-shot | 0.39 | 0.40 | 0.48 | 0.51 | 0.46 | 0.42 | 0.53 | 0.38 |
| 3-shot | 0.39 | 0.40 | 0.48 | 0.57 | 0.45 | 0.37 | 0.53 | 0.43 |
| 4-shot | 0.39 | 0.39 | 0.53 | 0.54 | 0.46 | 0.39 | 0.53 | 0.43 |
| 5-shot | 0.42 | 0.41 | 0.52 | 0.54 | 0.44 | 0.40 | 0.54 | 0.44 |

Table 5: Comparison of cloze prompting and our method with QK-Score. All experiments were ran on 4-optioned examples.

