# OpenReview forum: "Listening to the Wise Few: Select-and-Copy Attention Heads for Multiple-Choice QA"
_ICLR.cc/2025/Conference — Submitted to ICLR 2025_

### Official Review · Reviewer_U9do · 2024-10-18

**Soundness:** 3
**Presentation:** 3
**Contribution:** 3
**Rating:** 5
**Confidence:** 4

**Summary:**

The authors use a small amount of labeled data to identify "select-and-copy" heads. These are heads where information is being copied (via the attention mechanism) from a token the authors associate with a particular answer option (e.g., the newline after the option text) to the final token that will be used for prediction. This selection is primarily done by max "QK-score" (dot product between the query of the last token and key of the token associated with the answer option). The authors show "select-and-copy" heads are present in a variety of Llama models. They argue that using these heads for prediction leads to better accuracy and is less dependent than baseline on the order of answer options. The authors also run some design and head ablations, and explain an approach to finding "select-and-copy" heads in a label-free way.

**Strengths:**

* ORIGINALITY: I'm not very familiar with mechanistic interpretability work, but in my view this work seems quite novel. The authors find attention heads that seem to have a very well-defined function for MCQA, and show these are present in many models.
* QUALITY: The authors' experiments seem well-designed and their argument (at least regarding the presence of "select-and-copy" heads) is convincing. The authors also have a sizable appendix, suggesting they've tried a lot of things.
* CLARITY: The paper is mostly clear and easy to read.
* SIGNIFICANCE: I think this is significant in that it provides more insight into the mechanisms behind MCQA in LLMs.

**Weaknesses:**

* Primary weaknesses
  * This paper heavily emphasizes the zero-shot case, but I don't think it should be highlighted (I think it should be moved to the appendix if included at all). This is because in the zero-shot case the right way of answering (for the baseline models) is ambiguous. A human wouldn't know whether to respond with a letter vs the answer option text. I think e.g., Table 1, for example, should not show zero-shot results. I don't think the zero-shot setting is a fair setting for comparison.
  * I don't think the "E" and "F" options should be included (or at most this should be moved to appendix). As far as I know, adding the "E" and "F" options is not consistent with the majority of prior work in MCQA, and seems like an added variable that's not justified.
  * The authors pitch QK-score as being better than PriDe, and also having much improved accuracy across answer orders. However, I am not convinced of either of these. In the 1+ shot, no "E"/"F" setting PriDE seems as good or better. Also in e.g., Figure 3, the drop for PA is substantial for QK-score (just as substantial as for the alternatives). It seems like, from the appendix, the baseline is actually better than QK-score in many cases. Just to be clear, I don't think the authors' method needs to be more accurate than alternatives for the paper to be useful or accepted. I'm just saying the authors could maybe reassess their claims a little bit.
* The authors only consider Llama models, so it's unclear if these results apply to other LLMs.
* I don't fully follow the "Best Heads" part and Figure 5a in Section 6 despite having read it a few times. I definitely get the point being made, but I couldn't reproduce the result based on the description. To improve understanding and reproducibility, it might be nice to include a step-by-step description or pseudocode.
* I didn't take this into account in my rating, but I think the paper could benefit from another solid pass just for grammar. There are just enough errors that at times it was a bit distracting.
* See questions for more things I think could use clarification/improvement.

**Questions:**

* Is RoPE applied when using attention score?
* Why is "stochastic" used to imply "sums to one" on line 147 (I may be missing something)?
* For Llama base vs chat models was the same prompt used? Would this lead to worse performance for the baseline?
* Why the big difference in accuracy for e.g., HaluDialogue vs small difference in accuracy for MMLU? I don't find the argument on 322-323 convincing.
* Why is attention score included? It seems like QK-score is used as the default, and attention score is barely mentioned. My inclination would be to move the attention score parts to the appendix to prevent confusion over when which score is being used. At the very least, there could be more clarification on exactly when each is being used.
* In the unsupervised head finding part, what accuracies do the top heads achieve? I'm curious if they're like 90% as good as the best ones, or if they're much worse because their function matches but they're doing something entirely different.
* Why not ensemble heads?
* I'm curious why accuracy remains quite high (despite the drop) in the head removal ablation (especially in higher shot setting). Is it just that there are more than 10 "select-and-copy" heads? Or do "select-and-copy" heads only explain part of what's going on?

---

> ### Author Response · Authors · 2024-11-22
> **Response to Reviewer U9do (part 1: weaknesses)**
>
> On the main weaknesses:
>
> **(W1)** Thank you for your feedback on the zero-shot results. While we agree that baseline models may face ambiguity in response format (e.g., letter vs. option text), we included the zero-shot setting to highlight that the model's knowledge is captured in its layers. Specifically, the QK-score demonstrates comparable performance between zero-shot and few-shot settings, showing that the model possesses the knowledge even without external prompts and finetuning.
>
> We report both base and chat/instruct model results in Table 1 to provide a complete view of model capabilities. While we can move some zero-shot discussions to the appendix, we believe retaining some mention in the main text is essential for showcasing the model's fundamental abilities.
>
> It has been shown that few-shot introduces biases due to the selection and order of examples, which is why considering a zero-shot setup is also an important part of our research. Moreover, many modern models are fine-tuned on MCQA (multiple-choice question answering) during the SFT (supervised fine-tuning) stage, or relevant data is added during pretraining (see our common answer for all reviewers). This lets us tell that many models already have some degree of understanding of the format.
>
> **(W2)**  Including the "E" and "F" options addresses known biases in MCQA formats by filtering out models that tend to select the last option and by better aggregating uncertainty in predictions. This approach aligns with Ye et al. (2024), as we adopt their datasets and methodology, ensuring consistency with their framework while addressing these biases. Moreover, we show that the models rarely select these options, but they provide useful insights for head selection (you can look at Appendix B for more detailed information)
>
> **(W3)** Thank you for your insightful feedback. We will revise this section for clarity.
>
> The PRIDE method is designed to remove positional bias, while our method improves scores primarily by separating the generation process from the decision-making involved in option selection. Since positional bias is part of the decision-making, it is inherited by our method. On the other hand, the final model output integrates the decisions of multiple heads, some of which are more biased (see Figure 26 in the Appendix). We aim to select heads that rely on semantics rather than prior knowledge of answer distribution, which in some cases excludes the biased heads and reduces overall bias.
>
> We did not intend to position PriDE as a direct competitor but as complementary, with the potential for combination in future research. All these points will be clarified in our revised manuscript.
>
> Additionally, we acknowledge that QK-score is sometimes a bit worse than the baseline, particularly in larger models. We hypothesize that this may be due to limitations in head selection, which could be addressed in future work.
>
> **(W4)** Thank you for pointing this out! We have now addressed this concern by adding results on Qwen-2.5 and Phi-3.5-mini-instruct into our common response to reviewers for clarity and completeness. We will also add this result and results for other models to the Appendix of our paper.
>
> Besides, we will perform the experiments with larger models of the same families.
>
> **(W5)** We added a listing of Python code in Appendix D as an example of how best heads are calculated for 2 datasets shot-wise, and the procedure is the same for more datasets or when we want to calculate heads dataset-wise. After seeing the code, Figure 5A should become clearer as we coloured the best heads from the shot-mixed calculation and framed the best heads from the dataset-mixed calculation.
> This figure is intended to demonstrate that there are heads which are the best over most of datasets and most of setups simultaneously (both framed and dark-colored).
>
> **(W6)** We will ensure the paper undergoes another thorough round of proofreading to address any remaining grammatical issues and improve readability. Your patience and understanding are greatly appreciated.

---

> ### Author Response · Authors · 2024-11-22
> **Response to Reviewer U9do (part 2: questions)**
>
> Thank you for your interest in our work and your questions!
>
> > Is RoPE applied when using attention score?
>
> Yes, the attention matrices that LLaMA-2 outputs have RoPE already incorporated into their values.
>
> > Why is "stochastic" used to imply "sums to one" on line 147 (I may be missing something)?
>
> "Stochastic" is the commonly used definition for vectors whose elements are non-negative and “sums to one”. When referring to matrices, it means that matrix rows (or columns) are stochastic vectors. See, for example, R.A. Brualdi, S.V. Parter, H. Schneider, "The diagonal equivalence of a nonnegative matrix to a stochastic matrix" J. Math. Anal. Appl., page 2.
>
> > For Llama base vs chat models was the same prompt used? Would this lead to worse performance for the baseline?
>
> Yes, we used the same prompt. We tried to adjust the prompt design for instruct- and chat-tuned models, however, this didn’t bring a noticeable improvement in the baseline over the standard prompt in our experiments with LLaMA 2. We will try to investigate this further in the future.
>
> > Why the big difference in accuracy for e.g., HaluDialogue vs the small difference in accuracy for MMLU?
>
> It is somewhat hard to give a precise answer, but from what we can tell, questions from MMLU are aimed at the model's learned knowledge while questions in HaluDialogue are much more centered around relations between words (tokens). Previous works in the field showed that learned facts are mostly stored in fully connected layers of the transformer LLMs; therefore, the QK-score that operates on Queue and Key vectors of the input can’t add much to it (they, of course, have information from the previous fully-connected layer inside them). At the same time, different attention heads focus on different relations between tokens in the text and they do not contribute equally to the embeddings from the final layer of the model (that are then passed through the language modeling head). Therefore, the difference in accuracy may be caused by the fact that there are heads that focus on the right relationships between tokens but information from them in final embeddings is blurred by noise from other attention heads.
>
> Besides, we suspect that many modern models are fine-tuned on MMLU-like questions during the SFT (supervised fine-tuning) stage since it's a very important benchmark.
>
> > Why is attention score included?
>
> The attention score is close relative to our QK-score, but it is somewhat more intuitive. Thus we thought that our readers would like to see their comparison in the main text. The only difference between them is the integration of the RoPE component into attention and normalization.
>
> > In the unsupervised head-finding part, what accuracies do the top heads achieve?
>
> Heads (14, 20) and (14, 26) of the LLAMA-2-7B model are ranked as the top 2 by an unsupervised head-finding algorithm on four real datasets (see Figure 25, left part). Their accuracy results on these datasets are detailed in Figure 7. In contrast, the top-1 head on the synthetic dataset is (14, 4)  (see Figure 25, right part), which performs significantly worse. The heads (14, 20) and (14, 26) still appear again, but at second and third place. This suggests that the synthetic dataset may be a less effective choice for applying an unsupervised algorithm.
>
> > Why not ensemble heads?
>
> Thank you for this question. Our primary aim was to investigate the role of individual heads, but we will consider head ensembling for our future work.
>
> > I'm curious why accuracy remains quite high (despite the drop) in the head removal ablation (especially in higher shot settings).
>
> Thank you for your insightful observation. The relatively high accuracy in the head removal ablation, even with a drop, can be attributed to several factors. First, while the number of heads exceeds 10, we use a constant number of heads for comparison with random removal to maintain consistency. Additionally, as shown in Figure 10, we observe a trend where models with higher baseline accuracy tend to have more "good" heads, as evidenced by the comparison between LLaMA2-7B and LLaMA3-8B.
>
> To support this, we performed experiments on LLaMA2-7B to calculate the number of heads achieving accuracy greater than random (0.25) across various datasets and shot settings (0-5 shots):
>
> - MMLU: [47, 153, 181, 194, 204, 194]
> - CosmosQA: [65, 148, 172, 171, 174, 177]
> - HellaSwag: [33, 125, 148, 149, 151, 150]
> - HaluDialogue: [21, 97, 102, 128, 127, 117]
>
> The number of such heads stabilizes after 1-2 shots, but it is a bit different across datasets. However, we do not claim these heads are the sole mechanism helping the model solve MCQA tasks. To make such a claim, a more detailed circuit analysis would be required, which we leave for future work.

---

> ### Author Response · Authors · 2024-11-25
>
> Please respond to our post to let us know if the clarifications above suitably address your concerns about our work. We are happy to address any remaining points during the discussion phase; if the responses above are sufficient, we kindly ask that you consider raising your score.

---

> ### Author Response · Authors · 2024-12-02
>
> Dear Reviewer U9do,
>
> We thank you for your review and appreciate your time reviewing our paper.
>
> The end of the discussion period is close. We would be grateful if we could hear your feedback regarding our answers to the reviews. We are happy to address any remaining points during the remaining discussion period.
>
> Thanks in advance,
>
> Paper authors

---

### Official Review · Reviewer_nFPp · 2024-11-03

**Soundness:** 3
**Presentation:** 2
**Contribution:** 3
**Rating:** 6
**Confidence:** 3

**Summary:**

This work presents a new method for improving the evaluation of LLMs in MCQA by recognizing and using select-and-copy heads, which are particular attention heads. These attention heads consistently extract relevant information and improve response selection using the Query-Key Score (QK-score) and Attention Score. The strategy significantly improves MCQA benchmarks and a synthetic dataset for understanding. The study emphasizes the importance of intermediate attention states for disclosing underlying knowledge, particularly in smaller LLMs where typical output-based evaluation may understate the model's capabilities.

**Strengths:**

The paper introduces the concept of attention heads that are adept at copying information relevant to MCQA tasks, advancing the interpretability of LLMs.
QK-score and Attention Score are presented as innovative metrics that provide deeper insights into model decision-making processes.
Strong experimental setup with results across different models and settings enhances the credibility of the findings.

**Weaknesses:**

The approach focuses heavily on MCQA and may not generalize to open-ended or complex QA tasks.
Evaluating individual attention heads may be resource-intensive, especially for larger models.
While improving robustness, the paper does not fully address biases inherent to specific head selections.
Performance can differ based on head choice, potentially introducing instability in applications without careful selection.

**Questions:**

1. Investigate the relevance of the methodology to a broader range of QA formats and practical open-domain tasks.
2. Suggest ways or instruments that facilitate the selection and utilization of appropriate heads for enhanced adoption.
What precautions were implemented to prevent the identified select-and-copy heads from introducing unintentional biases in model outputs?
4. How is the effectiveness of these attention heads different for different model types, such as encoder-only vs. decoder-only?
5. Is it possible to scale cross-lingual or multilingual multiple-choice question answering evaluation?
6. How well do the QK-score and Attention Score work when used in models that have been fine-tuned for specific topic tasks?

---

> ### Author Response · Authors · 2024-11-25
> **Response to Reviewer nFPp - Part 1 (weaknesses)**
>
> Thank you for your thoughtful observations and questions. We would like to discuss the concerns you mentioned in your review:
>
> **(W1)** _The approach focuses heavily on MCQA and may not generalize to open-ended or complex QA tasks._
>
> We appreciate the reviewer’s thoughtful feedback.  We would like to clarify that the QK score is specifically designed to measure the selection and copying of information for answering questions. While it is not directly applicable to open QA, adapting this mechanism for such tasks could be a promising direction for future research.  Our study rather aims to uncover the model's hidden capabilities in answering MCQA beyond standard evaluations. The format of multiple-choice prompting is regular for many evaluation procedures [1].
>
> The QK method offers key insights, particularly advancing the understanding of the roles of attention heads [2] and the intrinsic mechanisms of MCQA [3]:
> - Enhanced Interpretability: The QK method identifies attention heads, particularly in middle layers, that use select-and-copy mechanisms to solve MCQA, providing insights into their role in reasoning.
> - Separating Format and Knowledge: Our approach distinguishes the model’s understanding of the MCQA format from its underlying knowledge. This is particularly evident in synthetic datasets, where the QK method achieves near-perfect results, unlike standard procedures.
> - Transformer Insights: The results demonstrate that select-and-copy attention heads consistently accumulate semantic meaning in query and key representations, shedding light on the internal workings of transformer models.
>
> [1] Leveraging Large Language Models for Multiple Choice Question Answering https://openreview.net/forum?id=yKbprarjc5B
>
> [2]Attention Heads of Large Language Models: A Survey  https://arxiv.org/pdf/2409.03752v2
>
> [3]Answer, Assemble, Ace: Understanding How LMs Answer Multiple Choice Questions, ICLR 2025 submission, https://openreview.net/forum?id=6NNA0MxhCH
>
> **(W2)**  _Evaluating individual attention heads may be resource-intensive, especially for larger models._
>
> Our approach focuses on calculating 4 to 6 scalar products between query and key vectors—one for each option—within the representations of a single selected attention head. This process does not introduce any computational overhead during the scoring of individual samples.
>
> The head selection process is performed once for the entire dataset, and its runtime is minimal. For example, it typically takes only 3–6 minutes on a Tesla V-100 GPU for the LLaMa 2-7B base model, to select the head and then use it without computational overhead. Furthermore, to simplify the process, we can utilize universal heads—selected from the best-performing heads across datasets—thereby eliminating the need for dataset-specific head selection. This alternative approach results in only a moderate reduction in accuracy, as demonstrated in Fig. 7 of our paper.
>
> Additionally, our method requires just a single inference run per sample and supports partial inference by bypassing higher layers. For instance, when using the universally best head for LLaMa 7B, which is located in the 15th layer, calculations for layers 16–32 can be skipped entirely, offering significant computational efficiency. We hope this clarifies our method and its advantages, and we welcome further feedback or questions.
>
> **(W3)** _While improving robustness, the paper does not fully address biases inherent to specific head selections. Performance can differ based on head choice, potentially introducing instability in applications without careful selection._
>
> Thank you for your thoughtful feedback and for raising this important point. Again, we would like to clarify that our primary goal is to develop a method for investigating transformer mechanisms rather than to design a tool specifically for application purposes. As such, our focus has been on understanding and interpreting the internal workings of models rather than on ensuring robustness for practical deployment.
>
> However, we acknowledge that head selection can introduce biases, and we have briefly explored this issue in Section 6 and Appendix J. Specifically, we examine potential biases associated with the selected heads, which could be related to the model's selection biases. However, a deeper investigation into this aspect is beyond the scope of the current work and remains an avenue for future research. We appreciate your insights, and we agree that addressing these biases more comprehensively is an important direction for further study.

---

> ### Author Response · Authors · 2024-11-25
> **Response to Reviewer nFPp - Part 2 (questions)**
>
> We would also like to discuss your questions:
>
> > **(Q1)** Investigate the relevance of the methodology to a broader range of QA formats and practical open-domain tasks.
>
> As we mentioned in answer to (W1), we see the adaptation of QK-score to openQA formats as a future direction. However, we have performed some experiments with cloze prompting QA format, for which we adapt QK-score. For example, in Standard Cloze prompting each answer choice is passed separately to the model and the final answer corresponds to the option on which the model had the highest probability of the first generated token. In such a setup we use QK-score as follows: we take the query score from the whole prompt and the key score only for the option content ( from `Answer:` and until the end).
>
> > **(Q2)** Suggest ways or instruments that facilitate the selection and utilization of appropriate heads for enhanced adoption. What precautions were implemented to prevent the identified select-and-copy heads from introducing unintentional biases in model outputs?
>
> We used the “E” and “F” options to filter out heads that exhibited uncertainty (e.g., those frequently choosing “I don’t know”) or a strong bias towards the last option (F). Additionally, throughout our paper, we often used a “permutation accuracy” score, which inherently penalizes heads that display excessive bias toward a particular option. Furthermore, we introduced an additional method for selecting heads that explicitly filters out those with a strong tendency to favour the same option, as well as those with generally low attention scores across options. Details on this method can be found in the section “Finding best heads without validation labels” (line 471).
> Finally, we identified heads showing good performance across many datasets, which makes them more reliable.
> In fact, our findings on unsupervised heads selection demonstrate that there indeed  exist heads which directly implement position bias. The heads we select for evaluation, in the contrary, are less biased among all.
>
> It’s important to note that the overall construction of our method avoids issues caused by output format misunderstanding, but does not necessarily cause the removal of all the biases. From the other hand, the existence of “stable” heads demonstrates that indeed some heads rely more on semantics of the answer rather that surface properties which are dataset-specific.
>
> > **(Q3)** How is the effectiveness of these attention heads different for different model types, such as encoder-only vs. decoder-only?
>
> Our main experiments are performed on decoder models. It is somewhat difficult to apply our method for Encoder-only models, because they are usually nor trained for Causal Language Modelling Task and most of them have very short context length preventing their usage with multiple in-context examples. We reformulated our prompts for the Masked Language Modelling task and explored several encoder-only models including RoBERTa (only in 0-shot setups) and Longformer. The accuracy of both baseline and QK-score methods were between 19% and 26% on all setups. Permutation Accuracy was below 10% in almost all cases.
>
> > **(Q4)** Is it possible to scale cross-lingual or multilingual multiple-choice question answering evaluation?
>
> This is an intriguing and complex question that generally falls outside the scope of this work. However, we conducted a few preliminary experiments with our synthetic dataset to explore this direction. Specifically, we created small Italian, French, and Russian versions of the dataset and tested our method on them. We found that the best heads for these versions closely overlap with those identified for the English version. Namely, we found that 7 out of top-10 best heads are shared across synthetic datasets on different languages for LLAMA-2-7B (including two heads that are also the best across our real datasets, i.e. (14, 20) and (14, 24)).
>
> Full results can be found in Appendix K.
>
> ---
>
> _Concluding remarks._ Please respond to our post to let us know if the clarifications above suitably address your concerns about our work. We are happy to address any remaining points during the discussion phase; if the responses above are sufficient, we kindly ask that you consider raising your score.

---

> ### Author Response · Authors · 2024-12-02
>
> Dear Reviewer nFPp,
>
> We thank you for your review and appreciate your time reviewing our paper.
>
> The end of the discussion period is close. We would be grateful if we could hear your feedback regarding our answers to the reviews. We are happy to address any remaining points during the remaining discussion period.
>
> Thanks in advance,
>
> Paper authors

---

### Official Review · Reviewer_4Bv1 · 2024-11-04

**Soundness:** 4
**Presentation:** 4
**Contribution:** 2
**Rating:** 5
**Confidence:** 5

**Summary:**

This work introduces two new metrics—the Query-Key Score (QK-score) and the Attention Score—that utilize select-and-copy attention heads within the models to better capture their underlying knowledge. The authors argue that relying solely on logit scores to select answers can be misleading, especially for smaller models struggling with rigid formats. By using intermediate attention representations, this method reveals deeper insights into the model’s understanding, yielding accuracy gains of up to 16% on MCQA benchmarks such as MMLU and HellaSwag. The study finds that middle-layer attention heads are particularly effective, whereas later layers tend to revise and diminish performance. Overall, this work contributes an approach that not only improves MCQA accuracy but also enhances interpretability of LLMs.

**Strengths:**

1. The authors introduce QK-score and Attention Score for deeper evaluation of LLMs.
2. This method yields significant gains in MCQA tasks, with up to 16% improvement on some benchmarks which is quite significant.
3. The work leverages internal attention heads, offering transparent answer selection.
4. The authors demonstrate the effectiveness of middle-layer attention heads over final layers.
5. This method is tested across models ranging from 7B to 70B parameters

**Weaknesses:**

1. While the experiments have been performed across different generations of llama models, showing generalization across model families could be important
2. Although the method is effective, there is complexity in terms of implementation. The applicability of the method to various practical scenarios remains questionable

**Questions:**

1. Some experiments / results on other model families
2. Comment on usability. (refer to comment #2 in weakness)
3. Given the complexity of the method, will be interesting to see latency analysis when compared to baseline.

---

> ### Author Response · Authors · 2024-11-24
> **Response to Reviewer 4Bv1 (part 1)**
>
> We thank the reviewer for the positive feedback. We will improve the presentation according to the suggestions. Below we address specific concerns one by one.
>
> __W1,Q1:__ _Other model families._
> __A:__ We applied our method to smaller models: Qwen 2.5-1.5B (-Instruct and -Base) and Phi-3.5-mini-Instruct (3.8B parameters). We identified attention heads within these models that utilize the select-and-copy mechanism to answer the questions, and assessed their capability in doing so.  The results are shown in the tables below.
>
> In particular, we confirmed that, remarkably, Qwen 2.5-1.5B-base also has several heads that are consistently good across real datasets and synthetic dataset, such as heads (20, 4) and (21, 11). The consistency of these good heads is similar to that observed in LLaMA-2-7B and other models in the LLaMA family. However, the layers with the best heads in Qwen 2.5 are closer to the final layer than those in the LLaMA-family models. Specifically, the best heads in this model are concentrated around layers 16-22, while the model has a total of 28 layers (see Appendix M “Best heads on synthetic dataset for Qwen 2.5-1.5B”). Nevertheless, the general pattern remains: very early and very late layers do not contain select-and-copy heads, and, in the middle layers, there are some consistently good heads across several datasets, as  stated in our paper.
>
>
> Metrics of our method applied to Qwen2.5-1.5B-Instruct:
>
> ```
> |                | MMLU            | Cosmos          | Hellaswag       | HaluDialogue    |
> |----------------|-----------------|-----------------|-----------------|-----------------|
> |                | 0-shot | 5-shot | 0-shot | 5-shot | 0-shot | 5-shot | 0-shot | 5-shot |
> | Baseline (acc) | 58.3   | 59.4   | 74.1   | 76.7   | 54.8   | 61.4   | 29.7   | 45.3   |
> | QK-score (acc) | 57.5   | 58.3   | 76.0   | 75.8   | 59.4   | 59.0   | 36.0   | 44.3   |
> | Baseline (PA)  | 49.2   | 50.2   | 67.8   | 70.8   | 46.2   | 52.4   | 19.3   | 34.2   |
> | QK-score (PA)  | 41.1   | 46.0   | 69.1   | 68.7   | 51.0   | 48.5   | 12.8   | 25.6   |
> ```
>
> Metrics of our method applied to Qwen2.5-1.5B-base:
>
> ```
> |                | MMLU            | Cosmos          | Hellaswag       | HaluDialogue    |
> |----------------|-----------------|-----------------|-----------------|-----------------|
> |                | 0-shot | 5-shot | 0-shot | 5-shot | 0-shot | 5-shot | 0-shot | 5-shot |
> | Baseline (acc) | 58.9   | 58.5   | 77.0   | 77.6   | 60.6   | 56.1   | 41.4   | 41.0   |
> | QK-score (acc) | 57.3   | 56.6   | 74.8   | 77.2   | 56.9   | 54.0   | 42.8   | 43.6   |
> | Baseline (PA)  | 49.1   | 47.7   | 70.1   | 71.7   | 49.8   | 43.3   | 29.0   | 29.7   |
> | QK-score (PA)  | 42.3   | 47.6   | 65.5   | 70.7   | 43.4   | 44.0   | 30.1   | 30.5   |
> ```
>
> Metrics of our method applied to Phi3.5-mini-instruct:
>
> ```
> |                | MMLU            | Cosmos          | Hellaswag       | HaluDialogue    |
> |----------------|-----------------|-----------------|-----------------|-----------------|
> |                | 0-shot | 5-shot | 0-shot | 5-shot | 0-shot | 5-shot | 0-shot | 5-shot |
> | Baseline (acc) | 65.9   | 68.1   | 77.5   | 81.1   | 78.0   | 75.7   | 59.7   | 58.2   |
> | QK-score (acc) | 66.1   | 65.8   | 81.8   | 81.6   | 78.4   | 75.1   | 63.2   | 65.5   |
> | Baseline (PA)  | 58.7   | 60.6   | 71.7   | 75.9   | 73.0   | 70.0   | 50.2   | 47.6   |
> | QK-score (PA)  | 56.2   | 57.5   | 75.5   | 74.9   | 71.7   | 69.0   | 52.2   | 52.7   |
> ```
>
> Regarding the proximity of the QK-score to the baseline score in these setups, we hypothesise that this may be because these models were more specifically fine-tuned for multiple-choice question answering (MCQA). This hypothesis is supported by the observation that these models (even relatively small ones with 1.5B parameters) achieve much better performance than LLaMA 2 with 7B parameters in baseline setups.
>
> Additionally, we observed that the performance of the baseline is very similar in both 0-shot and 5-shot setups for these three models, particularly for Qwen Base and Instruct. It seems that these models are so well-accustomed to the MCQA format that they do not even require example prompts to understand how to answer MCQA questions to the best of their ability. We hypothesise that this is connected with very effective propagation of the signal from the select-of-copy heads, which provides accuracy similar to the baseline scoring from the last layer.
>
> (continued in part 2)

---

> > ### Author Response · Authors · 2024-12-03
> > **To Reviewer 4Bv1**
> >
> > Dear Reviewer 4Bv1,
> >
> > Thank you for your time reviewing our paper. As the discussion period is coming to a close, we would greatly appreciate your feedback on our responses to your comments.
> > We have, in particular, demonstrated the generalization of our method across 4 additional model families including Qwen 2.5 (10 models, ranging from 1.5B to 72B) and others, please see the results above and in the appendices L,M and O.
> >
> > We understand the crucial role reviewers play in maintaining the quality of the conference, and your timely input would be very helpful.
> >
> > Thank you for your time and consideration.
> >  Best regards, Paper Authors

---

> ### Author Response · Authors · 2024-11-24
> **Response to Reviewer 4Bv1 (part 2)**
>
> (continued from part 1)
>
> __W2,Q2,Q3:__ _Complexity/latency/applicability/usability_
> __A:__ _Implementation complexity._
> The principal part of our method is the calculation of 4 or 6 scalar products between query and key vectors, one product per each option, in the representations of the single selected head. Our method does not lead to any computational overhead during each sample scoring. At the beginning, the head selection is run once for the whole dataset. It takes a few minutes depending on the dataset, e.g. 3-6 minutes for LlaMa 2 -7B base model on Tesla V-100. Moreover, we can use universal heads, from the set of the best heads across all the datasets, removing the need for the head selection with a moderate drop in accuracy (fig.7 in the paper). Our method requires a single inference run per sample. Interestingly, our method allows partial inference, ignoring significant part of higher layers. For example, if we use the universally best head  for LlaMa 7B, it  lies in the 15th layer, and there is no need to calculate layers 16 - 32.
>
> _Practical applicability._
> Our primary aim was to reveal and interpret the practical internal workings of the model. Namely, our score offers several insights, advancing, in particular, understanding of the roles of the attention heads, and of MCQA intrinsic mechanism.  Our method demonstrates which specific attention heads within the model use the select-and-copy mechanism capable of answering the given questions, and to what degree they are capable of doing so. This enhances interpretability by identifying which model components contribute to the model’s reasoning and through which mechanisms.  Our method helps also to separate the model’s understanding of the MCQA format from the model's actual underlying knowledge, as demonstrated in the synthetic dataset experiments where the answer is explicitly known.  Our results demonstrate also that the specific select-and-copy attention heads, whose list is remarkably similar across different datasets and several setups, accumulate the semantic meaning of phrases in the query and key representations of the phrases' last token. This sheds more light on the practical internal workings of transformer models.
>
> We believe that our approach, with its workable implementation complexity and contributions to model interpretability, is both practically applicable and valuable for advancing the field.
>
> _Concluding remarks._ Please respond to our post to let us know if the clarifications above suitably address your concerns about our work. We are happy to address any remaining points during the discussion phase; if the responses above are sufficient, we kindly ask that you consider raising your score.

---

### Official Review · Reviewer_jnc8 · 2024-11-04

**Soundness:** 2
**Presentation:** 4
**Contribution:** 3
**Rating:** 3
**Confidence:** 5

**Summary:**

The widely used evaluation for large language models, multiple-choice question answering (MCQA), is very brittle, especially for small models -- existing works show that even if models know the answer, it often cannot output the correct A/B/C/D due to all sorts of bias. This work proposes to tackle the problem by looking at a novel QK-score: they first select certain "select-and-copy" attention heads based on a validation set, and then calculate the query-key dot product between the option and the question (there are many possible ways, and the authors conducted thorough ablations).

The authors conducted comprehensive experiments on commonly used datasets, with zero-shot/many-shot experiments across model scales. The proposed method significantly improved over the standard MCQA baseline and some previously proposed methods. The analysis revealed interesting aspects, such as the meaning of a phrase is often encoded in the last token of the phrase.

My main concern is:

(1) Cloze completion has been widely used and has shown to be much more stable than MCQA in most standard evaluations. There is almost no discussion on it and also no empirical comparison. Since the work's main goal is to make evaluation more reliable, I found the lack of comparison significantly undermines this work's contribution.

(2) Improving the score doesn't make one evaluation better -- the authors should show that it reflects a better comparison that is more consistent with human evaluation or some intuition (for example, previous evaluations show much higher variance or reversed trends like an 80B model is worse than 7B; this new method fixed it).

**Strengths:**

(1) The brittleness of MCQA is well known and is a problem in evaluation. The proposed method is intuitive, simple, and effective.

(2) The authors conducted a comprehensive evaluation and interesting analysis that demonstrated the effectiveness of the method.

(3) The proposed method can be used beyond standard evaluation, especially in interpretability applications.

**Weaknesses:**

My main concern is as shown below

(1) Cloze completion has been widely used and has shown to be much more stable than MCQA in most standard evaluations. There is almost no discussion on it and also no empirical comparison. Since the work's main goal is to make evaluation more reliable, I found the lack of comparison significantly undermines this work's contribution.

(2) Improving the score doesn't make one evaluation better -- the authors should show that it reflects a better comparison that is more consistent with human evaluation or some intuition (for example, previous evaluations show much higher variance or reversed trends like an 80B model is worse than 7B; this new method fixed it).

**Questions:**

Please see the "weaknesses" section + the question below

(3) How does the variance of each method look like for the main table/figure, especially when sampling different in-context examples + different orders?

---

> ### Author Response · Authors · 2024-11-21
> **Response to Reviewer jnc8 (part 1)**
>
> We thank the reviewer for the constructive feedback and comments. We will improve the presentation according to the suggestions. Below we address specific comments one by one.
>
> __W1__: *Comparison with cloze completion.*
> __A__: Thank you for your feedback and the opportunity to clarify our contributions in relation to cloze completion. While cloze-style evaluation (cloze prompting) has been widely used for evaluating language models, it has certain drawbacks such as the "probability stealing" effect, where the correct answer's probability is spread across different surface forms [1,2]. It is also sensitive to prompt phrasing and may overfit to training patterns. Although MCQA prompting addresses some of these issues, it introduces its own biases, such as position and label bias, and is sensitive to sample order in few-shot settings, also models often struggle with the required output format [2,3].
>
> Our method addresses  several of the above issues by separating option selection from generation within the language model. Compared to cloze prompting and MCQA prompting, our approach is less sensitive to answer format and wording, and it reduces typical MCQA biases by ignoring the most biased attention heads. Due to these different biases, our method and cloze prompting can provide complementary insights. Also, our method requires only a single forward pass regardless of the number of answer choices, whereas cloze prompting requires an individual forward pass for each option, which may result in better computational efficiency for our method.
>
> We compared our method with cloze prompting on the LLaMA2-7B model. The results show that our method outperforms cloze prompting on the CosmosQA dataset in 2-, 3-, 4-, and 5-shot settings. On the MMLU dataset, our method yields results similar to cloze prompting. This demonstrates that our approach achieves comparable performance while offering complementary insights.
>
> &nbsp;&nbsp; __MMLU__
>
> |         |     Cloze |   QK     |
> |:--------|-------------:|---------:|
> | 0-shot  |         0.38 |     0.35 |
> | 1-shot  |         0.40 |     0.39 |
> | 2-shot  |         0.39 |     0.40 |
> | 3-shot  |         0.39 |     0.40 |
> | 4-shot  |         0.39 |     0.39 |
> | 5-shot  |         0.42 |     0.41 |
>
>
> &nbsp;&nbsp;   __CosmosQA__
>
> |         |     Cloze |   QK     |
> |:--------|-------------:|---------:|
> | 0-shot  |         0.49 |     0.46 |
> | 1-shot  |         0.51 |     0.50 |
> | 2-shot  |         0.48 |     0.51 |
> | 3-shot  |         0.48 |     0.57 |
> | 4-shot  |         0.53 |     0.54 |
> | 5-shot  |         0.52 |     0.54 |
>
> As large language models have advanced, the format of MCQA tasks has shifted from cloze prompting to multiple-choice formulations [4,5], which aligns with our focus, but we are  also adding  to our paper the above clarifications on the relation with cloze prompting.  By addressing the limitations of cloze prompting and  MCQA prompting, our method can contribute to more reliable and insightful model evaluation.
>
> [1] Increasing Probability Mass on Answer Choices Does Not Always Improve Accuracy, EMNLP 2023,
> [2] When Benchmarks are Targets: Revealing the Sensitivity of Large Language Model Leaderboards ACL 2024,
> [3] A Study on Large Language Models’ Limitations in Multiple-Choice Question Answering ICLR 2024,
> [4] OLMES: A Standard for Language Model Evaluations. arXiv:2406.08446,
> [5] OpenAI (2024). GPT-4 technical report. arXiv:2303.08774.
>
> (continued in part 2)

---

> ### Author Response · Authors · 2024-11-21
> **Response to Reviewer jnc8 (part 2)**
>
> (continued from the previous comment)
>
> __W2:__ _Improving the score doesn't make one evaluation better._
> __A:__ Our primary aim was not merely to achieve higher scores but to uncover the model's hidden potential capabilities in answering multiple-choice questions (MCQA), that the standard evaluation procedures do not reflect. With the QK method, we also seek to reveal and interpret the internal workings of the model. Namely, our score offers the following insights, advancing, in particular, understanding of the roles of the attention heads [2], and of MCQA intrinsic mechanism [1]:
>   * _Enhanced interpretability._ Our method demonstrates which specific attention heads within the model use the select-and-copy mechanism capable of answering the given questions, and to what degree they are capable of doing so. We show that specific attention heads in middle layers are more effective at solving MCQA tasks than the final unembedding layer. This enhances interpretability by identifying which model components contribute to the model’s reasoning and through which mechanisms.
>   * _Separation of format understanding and underlying knowledge._ Our method helps to separate the model’s understanding of the MCQA format from the model's actual underlying knowledge. This point is especially supported by our experiments on the synthetic dataset. While the model clearly "knows" the answers to these synthetic questions (as they are explicitly provided in the prompt), this knowledge is not apparent using standard MCQA procedures. In contrast, our method yields near-perfect results, aligning with the intuition that the model surely can solve this (very simple) task.
>   * _Uncovering other internal mechanisms  in transformers._ Our results demonstrate also that the specific select-and-copy attention heads, whose list is remarkably similar across different datasets and several setups, accumulate the semantic meaning of phrases in the query and key representations of the phrases' last token. This sheds more light on the internal workings of transformer models.
>
> [1]Answer, Assemble, Ace: Understanding How LMs Answer Multiple Choice Questions,  ICLR 2025 submission, https://openreview.net/forum?id=6NNA0MxhCH
> [2]Attention Heads of Large Language Models: A Survey  https://arxiv.org/pdf/2409.03752v2
>
> Please refer to the full list of our contributions at the end of section 1.
>
> __Q3:__ _How does the variance of each method look like for the main table?_
> __A:__ For LLaMA-2 7B, when sampling different in-context examples, the QK-score usually (in 70% of setups) has lower accuracy variance than the baseline; in the one-shot setup, this holds for all main datasets. For permutation accuracy, the QK-score also has lower variance in 65% of the cases. For example, for one-shot prompting on our datasets, we obtain the following standard deviations for sampling different in-context examples:
>
> | STD                  |  MMLU  | Cosmos | Hellaswag | HaluDialogue|
> |:--------------------:|:-----------:|:-----------:|:-----------:|:-----------:|
> | Baseline (acc)  | 0.0093 | 0.0346 | 0.0321 | 0.0193 |
> | QK-score (acc) | **0.0028** | **0.0152** | **0.0293** | **0.0189** |
> | Baseline (PA)   | 0.0103 | 0.0506 | **0.0335** | **0.0211** |
> | QK-score (PA)  | **0.0073** | **0.0386** | 0.0510 | 0.0266 |
>
> _Concluding remarks._ Please respond to our post to let us know if the clarifications above suitably address your concerns about our work. We are happy to address any remaining points during the discussion phase; if the responses above are sufficient, we kindly ask that you consider raising your score.

---

> > ### Comment · Reviewer_jnc8 · 2024-11-25
> > **Thanks for your response**
> >
> > Thanks for the new results! Now I agree that getting the best results out of a model can be a goal of the evaluation method. However, I am still not totally convinced by the first additional results you provided. I know it is a lot of ask, but it would be great if you can also provide the cloze vs. MCQA vs. yours comparison on the other two tasks, HellaSwag and Halu Dialogue.
> >
> > Also, I wonder what is the motivation behind choosing the four tasks. Why not use the tasks that OLMES picked (which are arguably more commonly used for benchmarking LLMs)? Again, I know it is a lot to ask for additional results at this point, but just want to hear your reasoning behind it for me to better understand. Ideally, you should show something like, if you add your method to OLMES, that will improve the model scores (max of cloze, MCQA, and yours).

---

> > > ### Author Response · Authors · 2024-12-03
> > > **To Reviewer  jnc8**
> > >
> > > Dear Reviewer  jnc8,
> > >
> > > Thank you for your time reviewing our paper.
> > > As the discussion period is coming to a close, we would greatly appreciate your feedback on our responses to your comments.
> > > We understand the crucial role reviewers play in maintaining the quality of the conference, and your timely input would be very helpful.
> > >
> > > Thank you for your time and consideration.
> > > Best regards,
> > > Paper Authors

---

> ### Author Response · Authors · 2024-11-28
> **Response to the Reviewer jnc8 comment**
>
> Thank you for the additional feedback that permits us to clarify further our approach.
>
> _Comparing with cloze methods._
> We appreciate your request for additional results. In response, we have included cloze-prompting experiments on HellaSwag and Halu Dialogue in Appendix O,   acknowledging that this may offer complementary insights. However, we would like to re-emphasize that our primary focus in this work is to uncover the inner decision mechanisms in large language models for multiple-choice question answering (MCQA). Therefore, the comparison between our QK score and the baseline methods requires  __token-wise identical prompts__, where all answer options are presented simultaneously.
>
> This approach ensures that we are examining the models' decision-making under consistent conditions. In contrast, cloze prompting employs a fundamentally __different type of prompt__ that does not include the answer options, leading to different model behaviors.
>
> Comparing cloze prompting with other methods can introduce variables that obscure the specific decision mechanisms we aim to study.  Answering without being distracted by the different answer options can be more easy or more difficult depending on the dataset and task structure. Analyzing the implications of this  on comparison of QK score with other prompting strategies is indeed a promising direction, but it is outside the scope of the present work.
>
> _Motivation behind choosing the four tasks._
> Our primary aim is to analyze the inner decision mechanisms of LLMs in multiple-choice settings. We selected four diverse tasks—MMLU, CosmosQA, HellaSwag, and Halu Dialogue—that are well-suited for probing various aspects of model reasoning in MCQA contexts. While OLMES uses commonly benchmarked tasks, our selection covered principal types of questions where  all answer options are presented simultaneously, which was sufficient for our purposes and aligns with our research objectives.
>
> _Integration with OLMES._
> We agree that integrating our method with OLMES is a promising direction. Due to time and space constraints, we could not include these results in the current work but consider this an interesting direction for future study.
>
> _Concluding remarks._ Please respond to our post to let us know if the clarifications above suitably address your concerns about our work. We are happy to address any remaining points during the discussion phase; if the responses above are sufficient, we kindly ask that you consider raising your score.

---

### Author Response · Authors · 2024-11-22
**Reply to the common concern about the generalisation capabilities of our method to models other than LLaMA 1-3, 7-70B**

We appreciate reviewers for comprehensive feedback! The main question that was expressed by the most reviewers is the generalisation capabilities of our method to model families other than LLaMA. Here we would like to address this concern.

We applied our method to smaller models: Qwen 2.5-1.5B (-Instruct and -Base) and Phi-3.5-mini-Instruct (3.8B parameters). We identified which attention heads within these models utilize the select-and-copy mechanism to answer the given questions and assessed their capability in doing so. Additionally, we found that the QK-scores for these heads are much closer to the baseline in terms of accuracy and permutation score, compared to the results observed for LLaMA 7B-70B. The results are shown in the tables below.

In particular, we confirmed that Qwen 2.5-1.5B-base has several heads that are consistently good across real datasets and synthetic dataset, such as heads (20, 4) and (21, 11). The consistency of these good heads is similar to that observed in LLaMA-2-7B and other models in the LLaMA family. However, the layers with the best heads in Qwen 2.5 are closer to the final layer than those in the LLaMA-family models. Specifically, the best heads in this model are concentrated around layers 16-22, while the model has a total of 28 layers (see Appendix M “Best heads on synthetic dataset for Qwen 2.5-1.5B”). Nevertheless, the general pattern still holds: very early and very late layers do not contain very strongly pronounced select-and-copy heads, and there are some consistently good heads across several datasets, as we indeed stated in our paper.

Metrics of our method applied to Qwen2.5-1.5B-Instruct:

```
|                | MMLU            | Cosmos          | Hellaswag       | HaluDialogue    |
|----------------|-----------------|-----------------|-----------------|-----------------|
|                | 0-shot | 5-shot | 0-shot | 5-shot | 0-shot | 5-shot | 0-shot | 5-shot |
| Baseline (acc) | 58.3   | 59.4   | 74.1   | 76.7   | 54.8   | 61.4   | 29.7   | 45.3   |
| QK-score (acc) | 57.5   | 58.3   | 76.0   | 75.8   | 59.4   | 59.0   | 36.0   | 44.3   |
| Baseline (PA)  | 49.2   | 50.2   | 67.8   | 70.8   | 46.2   | 52.4   | 19.3   | 34.2   |
| QK-score (PA)  | 41.1   | 46.0   | 69.1   | 68.7   | 51.0   | 48.5   | 12.8   | 25.6   |
```

Metrics of our method applied to Qwen2.5-1.5B-base:

```
|                | MMLU            | Cosmos          | Hellaswag       | HaluDialogue    |
|----------------|-----------------|-----------------|-----------------|-----------------|
|                | 0-shot | 5-shot | 0-shot | 5-shot | 0-shot | 5-shot | 0-shot | 5-shot |
| Baseline (acc) | 58.9   | 58.5   | 77.0   | 77.6   | 60.6   | 56.1   | 41.4   | 41.0   |
| QK-score (acc) | 57.3   | 56.6   | 74.8   | 77.2   | 56.9   | 54.0   | 42.8   | 43.6   |
| Baseline (PA)  | 49.1   | 47.7   | 70.1   | 71.7   | 49.8   | 43.3   | 29.0   | 29.7   |
| QK-score (PA)  | 42.3   | 47.6   | 65.5   | 70.7   | 43.4   | 44.0   | 30.1   | 30.5   |
```

Metrics of our method applied to Phi3.5-mini-instruct:

```
|                | MMLU            | Cosmos          | Hellaswag       | HaluDialogue    |
|----------------|-----------------|-----------------|-----------------|-----------------|
|                | 0-shot | 5-shot | 0-shot | 5-shot | 0-shot | 5-shot | 0-shot | 5-shot |
| Baseline (acc) | 65.9   | 68.1   | 77.5   | 81.1   | 78.0   | 75.7   | 59.7   | 58.2   |
| QK-score (acc) | 66.1   | 65.8   | 81.8   | 81.6   | 78.4   | 75.1   | 63.2   | 65.5   |
| Baseline (PA)  | 58.7   | 60.6   | 71.7   | 75.9   | 73.0   | 70.0   | 50.2   | 47.6   |
| QK-score (PA)  | 56.2   | 57.5   | 75.5   | 74.9   | 71.7   | 69.0   | 52.2   | 52.7   |
```

Regarding the proximity of the QK-score to the baseline score in these setups, we hypothesise that this may be because these models were more specifically fine-tuned for multiple-choice question answering (MCQA). This hypothesis is supported by the observation that these models (even relatively small ones with 1.5B parameters) achieve much better performance than LLaMA 2 with 7B parameters in baseline setups.

Additionally, we observed that the performance of the baseline is very similar in both 0-shot and 5-shot setups for these three models, particularly for Qwen Base and Instruct. It seems that these models are so well-accustomed to the MCQA format that they do not even require example prompts to understand how to answer MCQA questions to the best of their ability. We hypothesise that this is connected with very effective propagation of the signal from the select-of-copy heads, which provides accuracy similar to baseline scoring from the last layer.

We will include the complete results in all setups (including 1-, … 4- shot prompting) in the Appendix L of our paper before the rebuttal period ends. We also plan to add the results from bigger models.

---

> ### Author Response · Authors · 2024-11-26
> **Additional experiments on other model families**
>
> We applied our method to bigger models of Qwen family:
>
> Qwen 2.5-7B-base:
>
> ```
> |                | MMLU            | Cosmos          | Hellaswag       | HaluDialogue    |
> |----------------|-----------------|-----------------|-----------------|-----------------|
> |                | 0-shot | 5-shot | 0-shot | 5-shot | 0-shot | 5-shot | 0-shot | 5-shot |
> | Baseline (acc) | 67.0   | 70.1   | 81.9   | 85.2   | 82.0   | 85.2   | 54.8   | 60.7   |
> | QK-score (acc) | 67.0   | 70.4   | 86.9   | 87.7   | 81.5   | 84.8   | 62.9   | 65.1   |
> | Baseline (PA)  | 57.9   | 62.7   | 74.5   | 80.0   | 74.3   | 80.9   | 43.7   | 51.2   |
> | QK-score (PA)  | 59.2   | 62.8   | 82.7   | 84.1   | 73.9   | 80.2   | 50.6   | 52.7   |
> ```
>
> Qwen 2.5-7B-Instruct:
>
> ```
> |                | MMLU            | Cosmos          | Hellaswag       | HaluDialogue    |
> |----------------|-----------------|-----------------|-----------------|-----------------|
> |                | 0-shot | 5-shot | 0-shot | 5-shot | 0-shot | 5-shot | 0-shot | 5-shot |
> | Baseline (acc) | 56.7   | 68.8   | 68.5   | 85.7   | 75.8   | 81.7   | 40.3   | 66.7   |
> | QK-score (acc) | 68.0   | 70.3   | 85.0   | 87.0   | 79.9   | 82.9   | 56.1   | 72.0   |
> | Baseline (PA)  | 47.6   | 62.3   | 60.2   | 82.1   | 69.0   | 77.1   | 30.2   | 59.7   |
> | QK-score (PA)  | 60.5   | 62.2   | 80.1   | 82.3   | 74.2   | 78.3   | 44.3   | 62.4   |
> ```
>
> Qwen 2.5-14B-base:
>
> ```
> |                | MMLU            | Cosmos          | Hellaswag       | HaluDialogue    |
> |----------------|-----------------|-----------------|-----------------|-----------------|
> |                | 0-shot | 5-shot | 0-shot | 5-shot | 0-shot | 5-shot | 0-shot | 5-shot |
> | Baseline (acc) | 71.2   | 76.6   | 87.9   | 88.5   | 88.1   | 88.6   | 67.0   | 73.2   |
> | QK-score (acc) | 73.8   | 75.3   | 92.1   | 91.4   | 90.1   | 89.8   | 74.4   | 75.9   |
> | Baseline (PA)  | 61.8   | 70.3   | 82.5   | 84.0   | 83.2   | 84.5   | 56.8   | 64.9   |
> | QK-score (PA)  | 64.6   | 68.5   | 89.3   | 88.2   | 87.1   | 86.5   | 66.1   | 65.7   |
> ```
>
> Qwen 2.5-14B-Instruct:
>
> ```
> |                | MMLU            | Cosmos          | Hellaswag       | HaluDialogue    |
> |----------------|-----------------|-----------------|-----------------|-----------------|
> |                | 0-shot | 5-shot | 0-shot | 5-shot | 0-shot | 5-shot | 0-shot | 5-shot |
> | Baseline (acc) | 74.8   | 77.6   | 81.0   | 86.2   | 80.6   | 87.6   | 54.0   | 69.6   |
> | QK-score (acc) | 75.3   | 76.1   | 89.7   | 89.0   | 85.3   | 86.2   | 69.4   | 75.6   |
> | Baseline (PA)  | 66.8   | 72.4   | 75.0   | 82.6   | 75.1   | 84.6   | 45.5   | 63.5   |
> | QK-score (PA)  | 67.8   | 70.3   | 85.6   | 83.2   | 81.3   | 82.1   | 56.6   | 68.0   |
> ```
>
> For these models, the relative performance of our method versus the baseline across different shot configurations is similar to that observed for LLAMA-2 and LLAMA-3 models of the same sizes. Full results, including those for Dolly 3B-v2, Gemma 2B, and others, can be found in updated Appendix L.
>
> ---
>
> As the revision period is ending soon, we would greatly appreciate it if you could review our rebuttal and let us know whether it has addressed all of your concerns. Should you have any remaining points, we are more than willing to engage further during the discussion phase. If our responses have sufficiently resolved your concerns, we kindly ask you to consider revisiting your evaluation and adjusting your score accordingly.

---

### Meta-Review · Area_Chair_SLqz · 2024-12-13

**Metareview:**

This study introduces a novel method aimed at enhancing the evaluation of large language models (LLMs) in multiple-choice question answering (MCQA) by leveraging select-and-copy heads, which are specific attention heads. These heads consistently extract pertinent information, thereby improving response selection through the use of the Query-Key Score (QK-score) and Attention Score. The proposed strategy leads to significant advancements in MCQA benchmarks as well as on a synthetic dataset designed for comprehension.

However, the reviewers have raised significant concerns regarding the experimental setup, the generalizability of the proposed method, and its comparison with previous work. The authors should address these issues to enhance the paper's credibility and persuasiveness.

**Additional Comments On Reviewer Discussion:**

The reviewers have raised significant concerns regarding the experimental setup, the generalizability of the proposed method, and its comparison with previous work. The authors should address these issues to enhance the paper's credibility and persuasiveness.

---

### Decision · Program_Chairs · 2025-01-22

Reject